# Preference-Modulated Structural Attention for Multi-Objective Combinatorial Optimization

**Rongsheng Jia** [1 2]  **Jun Zhang** [1]  **Yifan Zhang** [2 3 4]  **Jian Cheng** [2 3 5]

## Abstract

Recent decomposition-based approaches have achieved significant success in Multi-Objective Combinatorial Optimization (MOCO). However, existing methods typically rely exclusively on node-centric representations, failing to capture the complementary representations provided by edge features for problem instances, resulting in a persistent optimality gap. To address this, we propose a preference-modulated structural attention mechanism to enhance problem representation by synergizing node and edge features. It includes: (1) Utilizing preference-modulated edge features as explicit structural biases during attention calculation, enabling model to perceive sub-problem structures conditioned on specific preferences, and (2) an edge feature aggregation strategy that dynamically incorporates node-specific context into edge representations to enhance the perception of preference-aware structures. Experiments on classic MOCOP benchmarks demonstrate the superiority of our approach in terms of both performance and generalization capabilities.

## 1. Introduction

Multi-Objective Combinatorial Optimization (MOCO) constitutes a fundamental challenge in operations research and computational intelligence, garnering extensive attention due to its theoretical complexity and widespread applicability (Liu et al., 2020). These problems are ubiquitous in critical real-world systems, ranging from logistics and transportation planning (Lust & Teghem, 2010; Jozefowiez et al., 2008) to smart manufacturing scheduling (Türkyılmaz et al., 2020) and strategic resource allocation (Bazgan et al., 2009). In such practical scenarios, decision-makers are compelled to navigate intricate trade-offs between mutually exclusive criteria, such as minimizing operational costs while maximizing service quality or fairness. Unlike Single-Objective Combinatorial Optimization (SOCO), which converges to a distinct global optimum, the intrinsic difficulty of MOCO lies in identifying the Pareto optimal solutions with different trade-offs among the objectives. Due to the NP-hard nature of MOCOPs, exact algorithms often become computationally intractable as the problem size expands, making them unsuitable for large-scale scenarios (Ehrgott et al., 2016). Consequently, heuristic methods have become the standard alternative. Nevertheless, these approaches typically require starting an iterative search from scratch for every new instance, leading to significant computational overhead. Moreover, traditional heuristics heavily depend on hand-crafted rules and domain-specific tuning, which restricts their ability to generalize across different types of MOCOPs (Blot et al., 2018).

Inspired by the substantial success of neural methods in solving Single-Objective Combinatorial Optimization (SOCO) problems (Kool et al., 2018; Kwon et al., 2020; Li et al., 2023a; Xiao et al., 2024; Chen & Tian, 2019; Li et al., 2023b; Zhou et al., 2023; Liu et al., 2024; Meng et al., 2025b;a), researchers have recently extended these techniques to the Multi-Objective Combinatorial Optimization (MOCO) domain. A predominant approach involves decomposing the intractable MOCO problem into a series of scalarized sub-problems. Specifically, each sub-problem is conditioned on a distinct preference vector and is solved in an end-to-end manner via Deep Reinforcement Learning (DRL). By aggregating the solutions from these sub-problems, the model can effectively approximate the Pareto front. Early neural approaches primarily relied on transfer learning (Li et al., 2020) or meta-learning (Chen et al., 2023a) to tackle decomposed sub-problems. These methods typically necessitate training separate models or performing extensive fine-tuning for each specific preference vector. Consequently, they are computationally resource-intensive and struggle to generalize to unseen preferences during inference. To ad-

---

[1]School of Mathematics and Statistics, Nanjing University of Science and Technology, Nanjing, China [2]C[2]DL, Institute of Automation, Chinese Academy of Sciences, Beijing, China [3]School of Artificial Intelligence, University of Chinese Academy of Sciences, Beijing, China [4]University of Chinese Academy of Sciences, Nanjing, China [5]AiRiA, Nanjing, China. Correspondence to: Jun Zhang <phil_zj@njust.edu.cn>, Yifan Zhang <yfzhang@nlpr.ia.an.cn>.

*Proceedings of the 43$^{rd}$ International Conference on Machine Learning*, Seoul, South Korea. PMLR 306, 2026. Copyright 2026 by the author(s).

dress these limitations, recent research has shifted towards a unified modeling strategy. By training a single preference-conditioned model to solve all sub-problems simultaneously, these approaches have achieved state-of-the-art performance across multiple MOCO benchmarks (Lin et al., 2022; Fan et al., 2024; Chen et al., 2025a).

The integration of node and edge information provides a richer representation for combinatorial optimization instances (Meng et al., 2025a). However, existing neural approaches for Multi-Objective Combinatorial Optimization (MOCO) predominantly rely on node features, largely neglecting the utilization of edge attributes. Consequently, these models fail to capture the critical complementarity between the global long-range dependencies facilitated by node-based mechanisms and the local topological structure encoded in edge features, thereby limiting their ability to fully perceive and reason about complex graph topologies. A novel paradigm involves leveraging Graph Convolutional Networks (GCNs) to extract edge features, coupled with a parallel encoding and dual-stream decoding architecture (Meng et al., 2025a). However, as the scale of sub-problems increases, equipping the model with heavy graph encoders exacerbates the growth of computational overhead. Furthermore, the over-smoothing phenomenon associated with deepening GCN layers limits their potential performance (Hussain et al., 2022). To address these challenges, we propose PMSA. As is shown in Figure 1, PMSA adopts a lightweight design by injecting preference-modulated edge features directly into the attention mechanism in the form of a structural bias. This strategy bypasses additional computational burdens while leveraging edge features as a structural prior, allowing the model to naturally perceive preference-specific distance characteristics. Furthermore, we facilitate the participation of node features in the edge aggregation process. This enables the dynamic update of edge representations during encoding, thereby enhancing the model's capability to perceive sub-problem structures. Consequently, the model effectively captures complementary structural information from edge features while maintaining low computational complexity. Our main contributions are summarized as follows:

- We propose PMSA, a lightweight neural solver for MOCO that utilizes a joint node-edge input mechanism.

- We devise a preference-modulated structural attention mechanism that explicitly injects edge attributes into attention scores as a prior to capture preference-specific topologies.

- We design a node-guided dynamic edge feature aggregation strategy where node features actively refine edge representations during encoding, enhancing the perception of sub-problem structures.

- Extensive evaluations on Bi-TSP and Bi-CVRP benchmarks confirm that our approach outperforms state-of-the-art baselines in both solution quality and inference efficiency.

## 2. Related Works

**Traditional Methods for MOCO.** Traditional methods for MOCO are generally categorized into two streams, exact methods and heuristic algorithms. Exact methods theoretically guarantee the identification of the precise Pareto front; however, as the problem scale increases, the solving time grows exponentially, rendering these algorithms intractable for large scale instances (Ehrgott et al., 2016; Bergman et al., 2022). Consequently, heuristic approaches—particularly Multi-Objective Evolutionary Algorithms (MOEAs) (Tian et al., 2021) have emerged as the primary alternative. Within this domain, the two most prominent paradigms are dominance-based MOEAs (Deb et al., 2002; Deng et al., 2022) and decomposition-based MOEAs (Zhou et al., 2012; Hu et al., 2024). Despite extensive research, the performance of these methods remains constrained by the need for complex hand-crafted designs and the substantial computational time required for convergence (Tian et al., 2021; Yi et al., 2020; Xie et al., 2022).

**Neural Methods for MOCO.** Benefiting from the tremendous success of Deep Reinforcement Learning (DRL) in Single-Objective Combinatorial Optimization Problems (SOCOPs), researchers have approached Multi-Objective Combinatorial Optimization (MOCO) by treating scalarized sub-problems as SOCOPs. Early approaches involved training separate networks for each scalarized sub-problem; however, maintaining specialized models for every sub-problem incurs high training overhead and maintenance difficulties (Li et al., 2020; Mossalam et al., 2016; Zhang et al., 2022). In recent years, researchers have shifted toward training a unified model to solve sub-problems with arbitrary preferences. For instance, (Lin et al., 2022) utilizes a hypernetwork to generate specific decoder parameters for each preference. More recent works have achieved superior performance by integrating preferences directly into the encoder (Chen et al., 2025a; Fan et al., 2024). Regarding orthogonal contributions, (Chen et al., 2023b) enhanced solution diversity by increasing the computational budget, while another work by (Chen et al., 2025b) augmented model capabilities by integrating both graph and image information of problem instances. (Fan et al., 2025) proposed routing sub-problems to distinct neural architectures and employed preference learning to reduce gradient variance during training.

**Edge-Node Method for NCO.** A limited number of studies have investigated the joint exploitation of node and edge attributes. (Joshi et al., 2019) designed a GCN-based predictor utilizing both coordinates and weights to direct beam search.

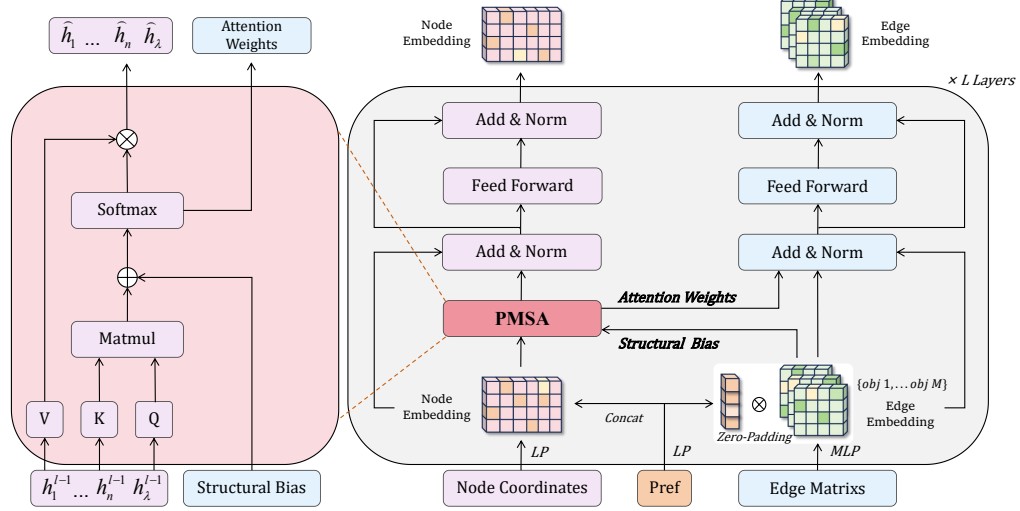

*Figure 1.* The Encoder architecture of our proposed preference-modulated structural attention model (PMSA).

(Zhou et al., 2024) proposed an instance-conditional model (ICAM) that fuses node and edge features to enhance generalization across varying problem scales. Wang et al. (Wang et al., 2025) introduced a distance-aware reshaping (DAR) technique that leverages Euclidean distances as biases. However, distinct from our approach, Wang et al. (Wang et al., 2025) primarily utilized edge features as a static structural bias within the decoding stage to mitigate attention dispersion in large-scale NCO. In contrast, our method integrates edge attributes during the encoding stage and explicitly incorporates a preference modulation mechanism, which is critical for balancing conflicting objectives. Furthermore, while the embedding of edge information has been successfully applied in Single-Objective Combinatorial Optimization Problems (SOCOPs), its application within the realm of Multi-Objective Combinatorial Optimization Problems (MOCOPs) remains scarce.

## 3. Preliminary

### 3.1. Multi-Objective Combinatorial Optimization

A Multi-Objective Combinatorial Optimization Problem (MOCOP) with $M$ optimization objectives is formally defined as:

$$\min_{x \in \mathcal{X}} \mathbf{F}(x) = (f_1(x), f_2(x), \ldots, f_M(x))^\top. \quad (1)$$

where $x$ denotes a solution within the discrete feasible set $\mathcal{X}$, and $f_i(x) : \mathcal{X} \to \mathbb{R}$ represents the $i$-th objective function to be minimized.

**Definition 3.1.** (Pareto Dominance) For two solutions $x, y \in \mathcal{X}$, solution $x$ is said to **dominate** solution $y$ (denoted as $x \prec y$) if and only if $f_i(x) \leq f_i(y)$ for all

$i \in \{1, \ldots, M\}$ and there exists at least one index $j$ such that $f_j(x) < f_j(y)$.

**Definition 3.2.** (Pareto Optimality) A solution $x^* \in \mathcal{X}$ is called Pareto optimal if there exists no other solution $x \in \mathcal{X}$ that dominates $x^*$ (i.e., $\nexists x \in \mathcal{X}$ such that $x \prec x^*$). The set of all Pareto optimal solutions constitutes the Pareto Set.

### 3.2. Decomposition-Based MOCO Method

To tackle the computational intractability of finding the exact Pareto set, we adopt a decomposition-based strategy. This approach transforms the original MOCOP into a set of scalarized sub-problems, each characterized by a specific preference vector $\boldsymbol{\lambda} = (\lambda_1, \ldots, \lambda_M)^T$, where $\sum_{m=1}^{M} \lambda_m = 1$ and $\lambda_m \geq 0$. For a given preference $\boldsymbol{\lambda}$, the scalarized objective function $g(x|\boldsymbol{\lambda})$ aggregates the multiple objective values into a single scalar value. A common scalarization method is the Weighted Sum (WS) approach, defined as:

$$g(x|\boldsymbol{\lambda}) = \sum_{m=1}^{M} \lambda_m f_m(x). \quad (2)$$

By solving a series of such sub-problems with diverse preference vectors sampled from a simplex, we can approximate the Pareto front effectively.

Following the POMO paradigm (Kwon et al., 2020), we model the solution construction as an MDP and train a unified stochastic policy $P_\theta(\pi|s, \boldsymbol{\lambda})$ to approximate the Pareto set. The policy generates a solution $\pi$ sequentially based on the problem instance $s$ and the preference vector $\boldsymbol{\lambda}$. We employ the REINFORCE (Williams, 1992) algorithm with a shared baseline to optimize the parameters $\theta$. The training

loss is defined as:

$$\mathcal{L}(\theta) = \frac{1}{N} \sum_{i=1}^{N} \left( g(\pi_i | \boldsymbol{\lambda}) - b(s, \boldsymbol{\lambda}) \right) \log P_\theta(\pi_i | s, \boldsymbol{\lambda}). \quad (3)$$

Here, $N$ denotes the number of diverse trajectories generated in parallel for instance $s$ by leveraging different starting nodes, where $\pi_i$ represents the $i$-th solution trajectory. The term $g(\pi_i | \boldsymbol{\lambda})$ corresponds to the scalarized objective cost of $\pi_i$ conditioned on preference $\boldsymbol{\lambda}$. The shared baseline $b(s, \boldsymbol{\lambda}) = \frac{1}{N} \sum_{j=1}^{N} g(\pi_j | \boldsymbol{\lambda})$ is derived from the average batch cost to effectively reduce gradient variance. For more details, see Appendix E.

## 4. Methodology

In this section, we present the integration of node and edge features. To avoid over-smoothing and high computational overhead of message-passing mechanism, we forgo the heavy graph encoders. This efficiency is particularly crucial for decomposition-based multi-objective frameworks, where the simultaneous processing of numerous sub-problems renders GCN-based approaches prohibitively expensive. In contrast, our proposed preference-modulated structural attention (PMSA) method cleverly circumvents the risk of smoothing associated with deep convolutions. We establish a synergistic evolutionary mechanism between nodes and edges. First, we utilize preference-modulated edge features as a structural bias to modulate attention scores. Second, we leverage normalized attention weights as explicit structural contexts to dynamically aggregate the relative dependency strengths between nodes into edge representations. our approach achieves SOTA performance.

### 4.1. Edge Feature Construction

This study encompasses classic benchmark problems widely accepted in MOCO literature, including the Multi-Objective Traveling Salesman Problem (Bi/Tri-TSP), the Multi-Objective Capacitated Vehicle Routing Problem (Bi-CVRP), and the Multi-Objective Knapsack Problem (Bi-KP) (see Appendix A). Tailored to the unique structural characteristics of each problem, we adopt differentiated initialization strategies for edge features. These raw features are subsequently projected into unified edge embeddings via a lightweight MLP.

**Bi/Tri-TSP.** These problems typically involve multiple independent cost metrics, where each objective corresponds to a distinct set of node coordinates. For the $m$-th optimization objective, we calculate the Euclidean distance between node $i$ and node $j$ to define the edge feature $e_{ij}^{(m)}$:

$$e_{ij}^{(m)} = \|\mathbf{x}_i^{(m)} - \mathbf{x}_j^{(m)}\|_2. \quad (4)$$

where $\mathbf{x}_i^{(m)}$ represents the coordinate vector of node $i$ for the $m$-th objective. This results in the construction of $M$ distinct distance matrices, explicitly characterizing the spatial transition costs across different objective dimensions.

**Bi-CVRP.** The two conflicting objectives in Bi-CVRP (total tour length and balance of routes) are generally defined on the same physical topology. Therefore, we construct a shared distance matrix based on the standard Euclidean distance using the single set of physical coordinates. This shared matrix serves as a common structural basis for both objectives.

**Bi-KP.** To integrate the Knapsack Problem into our unified neural framework, we abstract it as a node selection problem on a complete graph. Since KP lacks explicit geometric distances, we construct a value-based matrix derived from item attributes. Specifically, for the $m$-th objective, the raw feature for the edge connecting item $i$ and item $j$ is defined as the sum of their individual value-to-weight ratios:

$$e_{ij}^{(m)} = \text{Normalize} \left( \frac{v_i^{(m)}}{w_i} + \frac{v_j^{(m)}}{w_j} \right). \quad (5)$$

where $v$ and $w$ represent the item value and weight, respectively. This design intuitively aggregates the unit value density of both items, providing the model with critical heuristic information regarding the potential efficiency of selecting these items.

The raw edge features from $M$ objectives are projected into a unified embedding space compatible with the attention heads. Specifically, the edge embedding $\mathbf{h}_{ij}^e \in \mathbb{R}^H$ is computed via:

$$\mathbf{h}_e = \mathbf{W}_2 \sigma \left( \mathbf{W}_1 \left[ e_{ij}^{(1)}, \ldots, e_{ij}^{(M)} \right]^\top + \mathbf{b}_1 \right) + \mathbf{b}_2. \quad (6)$$

where $[\cdot]^\top$ denotes the concatenated input vector; $\mathbf{W}_1, \mathbf{b}_1$ and $\mathbf{W}_2, \mathbf{b}_2$ are the trainable parameters of the two linear layers; and $\sigma(\cdot) = \max(0, \cdot)$ denotes the ReLU activation function.

### 4.2. Preference-Modulated Structural Attention

Our model is built upon the prevalent Encoder-Decoder architecture and adopts POMO (Kwon et al., 2020) as the backbone. To equip the model with the capability to handle multi-objective sub-problems, we follow the WE-CA (Chen et al., 2025a) strategy for preference-aware node feature modulation. WE-CA utilizes the Feature-wise Linear Modulation (Perez et al., 2018) mechanism to inject preference information into node features via affine transformations, treating the preference vector as a special global token (see Appendix D). However, existing methods primarily focus

on feature modulation at the node level. We argue that optimal solutions under different preferences usually correspond to distinct graph topological structures, necessitating a structure-aware approach. To address this, we propose the preference-modulated structural attention (PMSA). We explicitly capture the preference-specific topology by projecting the preference vector and performing an element-wise multiplication with the edge feature $\mathbf{h}_{ij}^{e}$. This operation generates a preference-modulated structural bias $\mathbf{B}_{ij}$, defined as:

$$\mathbf{B}_{ij}(\boldsymbol{\lambda}) = \begin{cases} \mathbf{h}_{ij}^{e} \odot \mathrm{Linear}(\boldsymbol{\lambda}) & \text{if } i, j \leq N \\ 0 & \text{otherwise} \end{cases} \quad (7)$$

This bias $\mathbf{B}_{ij}$ not only integrates the underlying graph structure but also encodes the current optimization direction. Crucially, the condition $i, j \leq N$ implements a zero-padding strategy to align dimensions, ensuring that the structural bias is applied exclusively to edges between graph nodes and not to the global preference token. Subsequently, we inject this bias into the attention mechanism as an explicit structural prior. The final attention scores $A_{ij}$ are calculated by adding this bias to the scaled dot-product attention:

$$A_{ij} = \frac{(\mathbf{W}_Q \mathbf{h}_i)^{\top}(\mathbf{W}_K \mathbf{h}_j)}{\sqrt{d_k}} + \mathbf{B}_{ij}(\boldsymbol{\lambda}). \quad (8)$$

where $\mathbf{h}_i$ and $\mathbf{h}_j$ denote the input node features; $\mathbf{W}_Q, \mathbf{W}_K$ are query and key projection matrices; and $\odot$ represents the element-wise multiplication. In this manner, the attention mechanism can dynamically amplify or suppress the information flow along specific edges according to the current preference weights.

### 4.3. Edge Feature Aggregation

To enable edge features to better perceive sub-problem structures, we propose a dynamic edge update layer that allows structural embeddings to synchronize with node feature transitions. By leveraging the normalized attention score matrix as a real-time context, the edge features aggregate these interaction signals to update their own states. This mechanism effectively injects the current solving context into the edges, enabling the structural representations to shift adaptively as the solver focuses on different parts of the graph. This creates a feedback loop where edge features gain an enhanced capability to perceive and reinforce the sub-problem structure relevant to the current optimization step.

$$\mathbf{w}_{ij} = \mathbf{W}_e \mathrm{Softmax}\left(\mathcal{A}_{ij}^{(l)}\right) + \mathbf{b}_e. \quad (9)$$

$$\tilde{\mathbf{B}}_{ij}^{(l)}(\lambda) = \mathrm{Norm}\left(\mathbf{B}_{ij}^{(l)}(\lambda) + \mathbf{w}_{ij}\right). \quad (10)$$

$$\mathbf{B}_{ij}^{(l+1)}(\lambda) = \mathrm{Norm}\left(\tilde{\mathbf{B}}_{ij}^{(l)}(\lambda) + \mathrm{FFN}\left(\tilde{\mathbf{B}}_{ij}^{(l)}(\lambda)\right)\right). \quad (11)$$

## 5. Experiments

### 5.1. Experimental Settings

**Problems.** We conduct extensive experiments to evaluate the performance of PMSA across three representative MO-COPs: the Multi-Objective Traveling Salesman Problem (MOTSP) (Lust & Teghem, 2010), the Multi-Objective Capacitated Vehicle Routing Problem (MOCVRP) (Zajac & Huber, 2021), and the Multi-Objective Knapsack Problem (MOKP) (Ishibuchi et al., 2014). In MOTSP, the goal is to find a Hamiltonian cycle that simultaneously minimizes $M$ tour lengths calculated from independent coordinate sets. For MOCVRP, the objectives are to minimize the total traveling distance and the longest single route length for a fleet of vehicles serving customer demands. In MOKP, the aim is to maximize the total value across multiple objectives by selecting a subset of items, subject to a shared weight capacity constraint. We evaluate these models on standard problem sizes, setting $N \in \{20, 50, 100\}$ for MOTSP and MOCVRP, and $N \in \{50, 100, 200\}$ for MOKP to assess performance across varying scales.

**Hyperparameters.** Following standard implementation protocols in neural MOCO literature, we maintain consistent hyperparameter settings for fair comparison. We train the model for 200 epochs, where each epoch consists of 100,000 randomly generated instances processed with a batch size of 64. We employ the Adam optimizer (Kinga et al., 2015) with a learning rate of $1 \times 10^{-4}$ and a weight decay of $1 \times 10^{-6}$. The decomposition-based preference vectors are generated uniformly (Das & Dennis, 1998). Specifically, the population size is set to $N = 101$ for bi-objective optimization problems ($M = 2$) and $N = 105$ for tri-objective optimization problems ($M = 3$).

**Baselines.** To comprehensively evaluate the performance of PMSA, we compare it against three categories of prevalent baselines, all utilizing Weighted Sum (WS) scalarization for fair comparison. (1) Single-Model Methods: train a unified model to approximate the entire Pareto front, including PMOCO (Lin et al., 2022), CNH (Fan et al., 2024), and WE-CA (Chen et al., 2025a). Notably, CNH and WE-CA are implemented as cross-scale unified models, covering problem sizes $N \in [20, 100]$ for routing problems and $N \in [50, 200]$ for MOKP. (2) Multi-Model Methods: involve training or fine-tuning distinct models for sub-problems. This includes DRL-MOA (Li et al., 2020), which fully trains the initial model and utilizes parameter transfer to train subsequent models for fewer epochs, as well as MDRL (Zhang et al., 2022) and EMNH (Chen et al., 2023a), which fine-tune pretrained models for each specific decomposed sub-problem. (3) Non-Learnable Methods: encompass classic evolutionary algorithms and heuristics. We employ MOEA/D (Zhang & Li, 2007) and NSGA-II (running for 4,000 iterations) (Deb et al., 2002), MOGLS (4,000 iterations with 100 local

*Table 1.* Performance Comparisons on Bi-TSP and Bi-CVRP with 200 random instances.

| METHOD | BI-TSP20 | | | BI-TSP50 | | | BI-TSP100 | | |
|---|---|---|---|---|---|---|---|---|---|
| | HV (↑) | GAP (↓) | TIME (↓) | HV (↑) | GAP (↓) | TIME (↓) | HV (↑) | GAP (↓) | TIME (↓) |
| WS-LKH | 0.6270 | 0.00% | 10M | **0.6415** | **0.00%** | 1.8H | **0.7090** | **-0.10%** | 6H |
| MOEA/D | 0.6241 | 0.46% | 1.7H | 0.6316 | 1.54% | 1.8H | 0.6899 | 2.60% | 2.2H |
| NSGA-II | 0.6258 | 0.19% | 6.0H | 0.6120 | 4.60% | 6.1H | 0.6692 | 5.52% | 6.9H |
| MOGLS | **0.6279** | **-0.14%** | 1.6H | 0.6330 | 1.33% | 3.7H | 0.6854 | 3.23% | 11H |
| PPLS/D-C | 0.6256 | 0.22% | 26M | 0.6282 | 2.07% | 2.8H | 0.6844 | 3.37% | 11H |
| DRL-MOA | 0.6257 | 0.21% | 6S | 0.6360 | 0.86% | 9S | 0.6970 | 1.60% | 16S |
| MDRL | 0.6271 | -0.02% | 5S | 0.6364 | 0.80% | 8S | 0.6969 | 1.61% | 14S |
| EMNH | 0.6271 | -0.02% | 5S | 0.6364 | 0.80% | 8S | 0.6969 | 1.61% | 15S |
| PMOCO | 0.6259 | 0.18% | 6S | 0.6351 | 1.00% | 12S | 0.6957 | 1.78% | 26S |
| CNH | 0.6270 | 0.00% | 14S | 0.6387 | 0.44% | 17S | 0.7019 | 0.90% | 29S |
| WE-CA | 0.6270 | 0.00% | 7S | 0.6392 | 0.36% | 10S | 0.7034 | 0.69% | 21S |
| **PMSA** | 0.6272 | -0.03% | 8S | 0.6412 | 0.05% | 19S | 0.7070 | 0.18% | 38S |
| MDRL-AUG | 0.6271 | -0.02% | 34S | 0.6408 | 0.11% | 1.7M | 0.7022 | 0.86% | 14M |
| EMNH-AUG | 0.6271 | -0.02% | 34S | 0.6408 | 0.11% | 1.7M | 0.7023 | 0.85% | 14M |
| PMOCO-AUG | 0.6270 | 0.00% | 1.0M | 0.6395 | 0.31% | 3.2M | 0.7016 | 0.95% | 15M |
| CNH-AUG | 0.6271 | -0.02% | 1.5M | 0.6410 | 0.08% | 4.1M | 0.7054 | 0.41% | 16M |
| WE-CA-AUG | 0.6271 | -0.02% | 1.0M | 0.6413 | 0.03% | 3.3M | 0.7066 | 0.24% | 16M |
| **PMSA-AUG** | 0.6270 | 0.00% | 1.2M | **0.6415** | **0.00%** | 5.8M | 0.7083 | 0.00% | 22M |

| METHOD | BI-CVRP20 | | | BI-CVRP50 | | | BI-CVRP100 | | |
|---|---|---|---|---|---|---|---|---|---|
| | HV (↑) | GAP (↓) | TIME (↓) | HV (↑) | GAP (↓) | TIME (↓) | HV (↑) | GAP (↓) | TIME (↓) |
| MOEA/D | 0.4255 | 1.07% | 2.3H | 0.4000 | 2.61% | 2.9H | 0.3953 | 3.28% | 5.0H |
| NSGA-II | 0.4275 | 0.60% | 6.4H | 0.3896 | 5.14% | 8.8H | 0.3620 | 11.43% | 9.4H |
| MOGLS | 0.4278 | 0.53% | 9.0H | 0.3984 | 2.99% | 20H | 0.3875 | 5.19% | 72H |
| PPLS/D-C | 0.4287 | 0.33% | 1.6H | 0.4007 | 2.43% | 9.7H | 0.3946 | 3.45% | 38H |
| DRL-MOA | 0.4287 | 0.33% | 8S | 0.4076 | 0.75% | 12S | 0.4055 | 0.78% | 21S |
| MDRL | 0.4291 | 0.23% | 6S | 0.4082 | 0.61% | 13S | 0.4056 | 0.76% | 22S |
| EMNH | 0.4299 | 0.05% | 7S | 0.4098 | 0.22% | 12S | 0.4072 | 0.37% | 22S |
| PMOCO | 0.4267 | 0.79% | 6S | 0.4036 | 1.73% | 12S | 0.3913 | 4.26% | 22S |
| CNH | 0.4287 | 0.33% | 15S | 0.4087 | 0.49% | 17S | 0.4065 | 0.54% | 31S |
| WE-CA | 0.4290 | 0.26% | 7S | 0.4089 | 0.44% | 12S | 0.4068 | 0.46% | 26S |
| **PMSA** | 0.4293 | 0.19% | 8S | 0.4100 | 0.17% | 21S | 0.4079 | 0.20% | 45S |
| MDRL-AUG | 0.4294 | 0.16% | 12S | 0.4092 | 0.37% | 36S | 0.4072 | 0.37% | 2.8M |
| EMNH-AUG | **0.4302** | **-0.02%** | 12S | 0.4106 | 0.02% | 35S | 0.4079 | 0.20% | 2.8M |
| PMOCO-AUG | 0.4294 | 0.16% | 15S | 0.4088 | 0.46% | 36S | 0.3969 | 2.89% | 2.7M |
| CNH-AUG | 0.4299 | 0.05% | 22S | 0.4101 | 0.15% | 45S | 0.4077 | 0.24% | 2.5M |
| WE-CA-AUG | 0.4300 | 0.02% | 14S | 0.4103 | 0.10% | 40S | 0.4081 | 0.15% | 2.5M |
| **PMSA-AUG** | 0.4301 | 0.00% | 16S | **0.4107** | **0.00%** | 1.2M | **0.4087** | **0.00%** | 4.6M |

search steps) (Jaszkiewicz, 2002), and PPLS/D-C (200 iterations) (Shi et al., 2022). These methods utilize 2-opt heuristics for MOTSP/MOCVRP and greedy transformations for MOKP. Additionally, we utilize powerful solvers combined with WS scalarization, specifically WS-LKH (Helsgaun, 2000) for MOTSP and WS-DP for MOKP.

**Metrics.** We employ the widely used Hypervolume (HV) (While et al., 2006) metric to evaluate the performance of MOCO methods (see Appendix C), where a higher HV value indicates a superior solution set. We report the average HV over 200 test instances, along with the gap to the reference Pareto front (Gap) and the total inference time. Methods with the suffix "-AUG" denote results obtained using instance augmentation to further enhance performance (Lin et al., 2022) (see Appendix B). To assess statistical significance, we conduct the Wilcoxon rank-sum test with a significance level of 1% (Wilcoxon, 1992). The best results are highlighted in bold, and the second-best results are underlined. All experiments are implemented in Python and executed on a server equipped with an NVIDIA TITAN RTX 3090 GPU and an Intel Xeon Gold 5220 CPU.

*Table 2.* Performance Comparisons on Bi-KP and Tri-TSP with 200 random instances.

| METHOD | BI-KP50 HV (↑) | GAP (↓) | TIME (↓) | BI-KP100 HV (↑) | GAP (↓) | TIME (↓) | BI-KP200 HV (↑) | GAP (↓) | TIME (↓) |
|---|---|---|---|---|---|---|---|---|---|
| WS-DP | **0.3561** | **0.00%** | 22M | 0.4532 | 0.00% | 2.0H | 0.3601 | 0.03% | 5.8H |
| MOEA/D | 0.3540 | 0.59% | 1.6H | 0.4508 | 0.53% | 1.7H | 0.3581 | 0.58% | 1.8H |
| NSGA-II | 0.3547 | 0.39% | 7.8H | 0.4520 | 0.26% | 8.0H | 0.3590 | 0.33% | 8.4H |
| MOGLS | 0.3540 | 0.59% | 5.8M | 0.4510 | 0.49% | 10H | 0.3582 | 0.56% | 18H |
| PPLS/D-C | 0.3528 | 0.93% | 18M | 0.4480 | 1.15% | 47M | 0.3541 | 1.69% | 1.5H |
| DRL-MOA | 0.3559 | 0.06% | 9s | 0.4531 | 0.02% | 18s | 0.3601 | 0.03% | 1.0M |
| MDRL | 0.3530 | 0.87% | 6s | 0.4532 | 0.00% | 20s | 0.3601 | 0.03% | 1.2M |
| EMNH | **0.3561** | **0.00%** | 6s | **0.4535** | **-0.07%** | 20s | **0.3603** | **-0.03%** | 1.2M |
| PMOCO | 0.3552 | 0.25% | 9s | 0.4523 | 0.20% | 19s | 0.3595 | 0.19% | 1.3M |
| CNH | 0.3556 | 0.14% | 18s | 0.4527 | 0.11% | 27s | 0.3598 | 0.11% | 1.2M |
| WE-CA | 0.3558 | 0.08% | 9s | 0.4531 | 0.02% | 21s | 0.3602 | **0.00%** | 1.1M |
| **PMSA** | **0.3561** | **0.00%** | 16s | 0.4532 | 0.00% | 35s | 0.3602 | **0.00%** | 1.9M |

| METHOD | TRI-TSP20 HV (↑) | GAP (↓) | TIME (↓) | TRI-TSP50 HV (↑) | GAP (↓) | TIME (↓) | TRI-TSP100 HV (↑) | GAP (↓) | TIME (↓) |
|---|---|---|---|---|---|---|---|---|---|
| WS-LKH | **0.4712** | **0.00%** | 12M | **0.4440** | **-0.02%** | 1.9H | **0.5076** | **-0.16%** | 6.6H |
| MOEA/D | 0.4702 | 0.21% | 1.9H | 0.4314 | 2.82% | 2.2H | 0.4511 | 10.99% | 2.4H |
| NSGA-II | 0.4238 | 10.06% | 7.1H | 0.2858 | 35.62% | 7.5H | 0.2824 | 44.28% | 9.0H |
| MOGLS | 0.4701 | 0.23% | 1.5H | 0.4211 | 5.14% | 4.1H | 0.4254 | 16.06% | 13H |
| PPLS/D-C | 0.4698 | 0.30% | 1.4H | 0.4174 | 5.97% | 3.9H | 0.4376 | 13.65% | 14H |
| DRL-MOA | 0.4699 | 0.28% | 6s | 0.4303 | 3.06% | 9s | 0.4806 | 5.17% | 19s |
| MDRL | 0.4699 | 0.28% | 5s | 0.4317 | 2.75% | 9s | 0.4852 | 4.26% | 16s |
| EMNH | 0.4699 | 0.28% | 5s | 0.4324 | 2.59% | 9s | 0.4866 | 3.99% | 16s |
| PMOCO | 0.4693 | 0.40% | 6s | 0.4315 | 2.79% | 8s | 0.4858 | 4.14% | 18s |
| CNH | 0.4698 | 0.30% | 10s | 0.4358 | 1.82% | 14s | 0.4931 | 2.70% | 26s |
| WE-CA | 0.4707 | 0.11% | 5s | 0.4389 | 1.13% | 9s | 0.4985 | 1.64% | 20s |
| **PMSA** | **0.4712** | **0.00%** | 6s | 0.4430 | 0.20% | 16s | 0.5044 | 0.47% | 36s |
| MDRL-AUG | **0.4712** | **0.00%** | 2.6M | 0.4408 | 0.70% | 25M | 0.4958 | 2.17% | 1.7H |
| EMNH-AUG | **0.4712** | **0.00%** | 2.6M | 0.4418 | 0.47% | 25M | 0.4973 | 1.87% | 1.7H |
| PMOCO-AUG | **0.4712** | **0.00%** | 5.1M | 0.4409 | 0.68% | 28M | 0.4956 | 2.21% | 1.7H |
| CNH-AUG | 0.4704 | 0.17% | 33M | 0.4409 | 0.68% | 33M | 0.4996 | 1.42% | 2.1H |
| WE-CA-AUG | **0.4712** | **0.00%** | 31M | 0.4432 | 0.16% | 31M | 0.5035 | 0.65% | 1.8H |
| **PMSA-AUG** | **0.4712** | **0.00%** | 33M | 0.4439 | 0.00% | 55M | 0.5068 | 0.00% | 3.2H |

## 5.2. Experimental Results

**Comparison Analysis.** Tables 1 and 2 present the comparative results between our unified model and various MOCO baselines. PMSA consistently outperforms WE-CA across nearly all benchmarks. With the introduction of instance augmentation, our model achieves further performance gains. Especially, in MOTSP tasks, our model without instance augmentation surpasses all baselines equipped with instance augmentation, significantly narrowing the optimality gap. For instance, on Tri-TSP100, PMSA achieves a Hypervolume of 0.5044 compared to 0.5035 for WE-CA-Aug. This represents a 180 reduction in solving time while surpassing the best performance of the WE-CA model. While traditional non-learnable solvers require extensive computation time (e.g., WS-LKH takes 6 hours for Bi-TSP), PMSA achieves an optimality gap of 0.28% in just 38 sec-

onds, and reaches a 0.1% gap within 22 minutes with instance augmentation. Experimental results demonstrate that PMSA yields modest enhancements at the $N = 20$ scale, yet delivers increasingly substantial performance gains on larger instances. This scalability is driven by a key insight: as the instance size expands, standard attention weights inevitably diffuse over a broader neighborhood. To mitigate this dilution, our proposed edge flow mechanism acts as a structural anchor that effectively calibrates the attention flow. By reinforcing the focus on high-priority neighbors and suppressing long-range noise, this mechanism prevents attention over-dispersion. Consequently, PMSA preserves critical structural integrity, unlocking significant performance breakthroughs in large-scale and highly complex problem settings.

**Experimental Results on Larger-Scale Instances.** To as-

*Table 3.* Performance comparison on larger-scale instances.

| | BI-TSP-150 | | | BI-TSP-200 | | |
|---|---|---|---|---|---|---|
| METHOD | HV ($\uparrow$) | GAP ($\downarrow$) | TIME | HV ($\uparrow$) | GAP ($\downarrow$) | TIME |
| **WS-LKH** | **0.7149** | **-0.95%** | 13H | **0.7490** | **-0.81%** | 22H |
| NSGA-II | 0.6659 | 5.97% | 6.8H | 0.7045 | 5.18% | 6.9H |
| MOEA/D | 0.6809 | 3.85% | 2.4H | 0.7139 | 3.92% | 2.7H |
| MOGLS | 0.6768 | 4.43% | 22H | 0.7114 | 4.25% | 38H |
| PPLS/D-C | 0.6784 | 4.21% | 21H | 0.7106 | 4.36% | 32H |
| DRL-MOA | 0.6901 | 2.56% | 45S | 0.7219 | 2.84% | 1.5M |
| PMOCO | 0.6910 | 2.43% | 50S | 0.7231 | 2.68% | 1.5M |
| MDRL | 0.6922 | 2.26% | 40S | 0.7251 | 2.41% | 1.4M |
| EMNH | 0.6930 | 2.15% | 40S | 0.7260 | 2.29% | 1.4M |
| CNH | 0.6985 | 1.37% | 1.1M | 0.7292 | 1.86% | 1.9M |
| WE-CA | 0.7008 | 1.04% | 57S | 0.7346 | 1.13% | 1.6M |
| **PMSA** | 0.7062 | 0.28% | 1.7M | 0.7408 | 0.30% | 2.8M |
| PMOCO-AUG | 0.6967 | 1.62% | 47M | 0.7283 | 1.98% | 1.6H |
| MDRL-AUG | 0.6976 | 1.50% | 47M | 0.7299 | 1.76% | 1.6H |
| EMNH-AUG | 0.6983 | 1.40% | 47M | 0.7307 | 1.66% | 1.6H |
| CNH-AUG | 0.7025 | 0.80% | 52M | 0.7307 | 1.66% | 1.7H |
| WE-CA-AUG | 0.7044 | 0.54% | 50M | 0.7381 | 0.66% | 1.7H |
| **PMSA-AUG** | 0.7082 | 0.00% | 1.5H | 0.7430 | 0.00% | 2.9H |

*Table 4.* Performance on specific preference subproblems on Bi-TSP100 with 200 random instances .

| Method | $\boldsymbol{\lambda} = (1, 0)$ | | $\boldsymbol{\lambda} = (0.5, 0.5)$ | | $\boldsymbol{\lambda} = (0, 1)$ | |
|---|---|---|---|---|---|---|
| | **Obj** ($\downarrow$) | **Gap** ($\downarrow$) | **Obj** ($\downarrow$) | **Gap** ($\downarrow$) | **Obj** ($\downarrow$) | **Gap** ($\downarrow$) |
| LKH | 7.7632 | 0.00% | 17.3094 | 0.00% | 7.7413 | 0.00% |
| POMO-Aug | 7.7659 | 0.03% | 17.4421 | 0.77% | 7.7716 | 0.39% |
| MDRL-Aug | 8.0316 | 3.46% | 17.6209 | 1.80% | 8.0290 | 3.72% |
| EMNH-Aug | 8.0620 | 3.85% | 17.5979 | 1.67% | 8.0439 | 3.91% |
| PMOCO-Aug | 8.1401 | 4.85% | 17.5723 | 1.52% | 8.1202 | 4.89% |
| WE-CA-Aug | 7.8198 | 0.73% | 17.4633 | 0.89% | 7.7937 | 0.68% |
| **PMSA-Aug** | 7.7896 | 0.34% | 17.3516 | 0.24% | 7.7671 | 0.33% |

sess the cross-scale generalization capability of our model, we evaluate PMSA on larger problem instances, specifically Bi-TSP150 and Bi-TSP200. As shown in Table 3, PMSA outperforms all neural baselines by a significant margin, consistently demonstrating superior generalization performance across varying problem scales.

**Generalization to Out-of-Distribution.** We evaluate the out of distribution generalization capability of our model on benchmark instances by distinct distribution. Figure 2 illustrates the performance of all comparison methods on Bi-TSP instances from TSPLIB (KroAB100, KroAB150, KroAB200). The experimental results demonstrate that PMSA consistently exhibits superior OOD generalization performance compared to other neural baselines across the evaluated instances. Complete benchmark results are detailed in Appendix F.

**Performance on Specific Preference Subproblems.** As shown in Table 4, earlier neural multi-objective approaches struggle to converge to the Pareto front, exhibiting no-

ticeable optimality gaps ($> 1.5\%$) across all subproblems. While the single-objective specialist POMO-Aug performs exceptionally well on extreme preferences, its performance degrades on the challenging balanced scenario ($\boldsymbol{\lambda} = (0.5, 0.5)$). In contrast, our PMSA-Aug delivers the most robust performance. It consistently outperforms the strongest MOCO baseline, WE-CA-Aug, on all subproblems and achieves a significantly lower gap than POMO-Aug on the balanced weights (0.24% vs. 0.77%), demonstrating the superiority of the proposed structural preference adaptation in handling diverse trade-offs.

**Ablation Study.** To verify the effectiveness of the key components in PMSA, we conduct ablation studies on Bi-TSP instances with $N = 100$ and $N = 200$. We compare the full model against two variants: (1) w/o Pref-Mod, which removes the preference injection operation from the edge features, resulting in a structural bias that is agnostic to the current optimization direction; and (2) w/o Edge-Agg, which eliminates the dynamic edge aggregation mechanism, keeping the edge embeddings static throughout the encoding

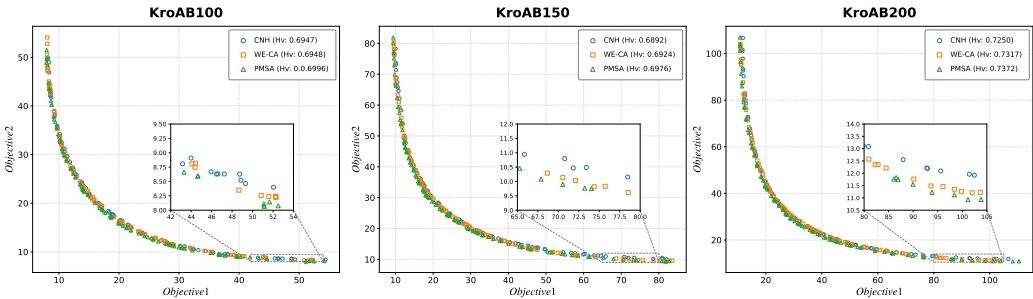

*Figure 2.* Pareto fronts on TSPLIB benchmark instances.

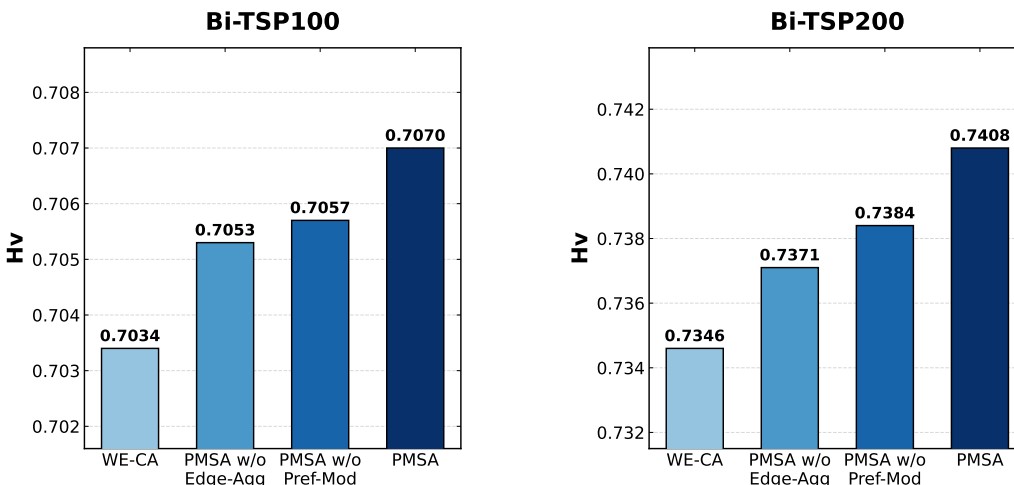

*Figure 3.* Ablation study on Bi-TSP.

layers. As shown in Figure 3, the full PMSA model consistently outperforms both variants. The performance drop in w/o Pref-Mod validates our core hypothesis that optimal solutions under different preferences require distinct topological structures, while the inferiority of w/o Edge-Agg demonstrates the necessity of iteratively refining structural information to capture complex edge dependencies.

## 6. Conclusion

In this paper, we propose PMSA, a method that synergizes node-based attention mechanisms with preference-modulated edge feature embeddings. The former captures long-range dependencies between nodes via a global receptive field, while the latter focuses on the fine-grained characterization of local topological structures. This organic complementarity of global vision and local perception significantly enhances the model's capability to represent sub-problem graph structures. Furthermore, we design an edge aggregation strategy that utilizes normalized attention scores as context to guide the dynamic edge update process, thereby enabling more precise perception of sub-problem structures under specific preferences. Extensive experiments

on benchmarks demonstrate that our method outperforms state-of-the-art neural methods. Ablation studies further underscore the significance of edge feature injection, edge feature aggregation, and preference injection in achieving superior performance.

Despite the promising results, there are limitations that merit further investigation. The current approach relies on pre-defined preference weights, which may lack flexibility when handling irregular Pareto fronts. Designing an adaptive preference mechanism to dynamically adjust search directions remains a crucial direction. Our future work will focus on improving the model's generalizability to large-scale real-world MOCOPs with complex constraints, exploring how to extend this framework to more challenging practical application scenarios.

## Impact Statement

This work presented in this paper aims to advance research in the field of machine learning, drive industrial development, and enhance decision-making capabilities. We believe our work holds potential for positive societal and practical impacts.

## Acknowledgements

This work was supported in part by the National Key R&D Program of China (No. 2025ZD0122000), the Strategic Priority Research Program of Chinese Academy of Sciences (XDA0480203), NSFC 62273347, the Key Research and Development Program of Jiangsu Province (BE2023016).

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

# A. Formalization of MOCO Problems

In this section, we provide the formal definitions and the data generation protocols for the three Multi-Objective Combinatorial Optimization problems studied in this work: the Multi-Objective Traveling Salesman Problem (MOTSP), the Multi-Objective Capacitated Vehicle Routing Problem (MOCVRP), and the Multi-Objective Knapsack Problem (MOKP).

### A.1. Multi-Objective Traveling Salesman Problem (MOTSP)

The MOTSP is defined on a complete graph $\mathcal{G} = (\mathcal{V}, \mathcal{E})$, where $\mathcal{V} = \{1, \ldots, N\}$ represents the set of cities. Each edge $(i, j) \in \mathcal{E}$ is associated with $m$ distinct costs $\mathbf{c}_{ij} \in \mathbb{R}^m$. The objective is to find a Hamiltonian cycle $\pi$ that minimizes the sum of costs for each objective:

$$\min_{\pi} \mathbf{F}(\pi) = (f_1(\pi), f_2(\pi), \ldots, f_M(\pi)), \text{ where } f_m(\pi) = \sum_{i=1}^{N-1} c_{\pi_i, \pi_{i+1}}^{(m)} + c_{\pi_N, \pi_1}^{(m)}, \quad \forall m \in \{1, \ldots, M\}. \quad (12)$$

**Data Generation.** Following standard conventions in recent learning-based MOCO studies (Lin et al., 2022), we generate instances where node locations are sampled uniformly from the unit square $[0, 1]^2$. To establish the necessary trade-offs among $M$ objectives, we generate $M$ independent sets of node coordinates, consistently employing the Euclidean distance as the cost metric for each objective.

### A.2. Multi-Objective Capacitated Vehicle Routing Problem (MOCVRP)

In the Multi-Objective Capacitated Vehicle Routing Problem (MOCVRP), the objective is to simultaneously optimize two conflicting goals for a fleet of vehicles with capacity $Q$ serving a depot and $N$ customer nodes, minimizing the total traveling distance, and minimizing the length of the longest single route. Each customer $i$ is characterized by a coordinate $\mathbf{x}_i$ and a demand $d_i$, and a feasible solution $\pi$ consists of a set of routes $\{\pi^{(1)}, \ldots, \pi^{(K)}\}$ that start and end at the depot, ensuring the total demand of each route does not exceed $Q$. We generate instances by sampling depot and customer coordinates uniformly from $[0, 1]^2$ and customer demands uniformly from the discrete set $\{1, \ldots, 9\}$. The vehicle capacity is scaled according to problem size, with $Q$ set to 30, 40, and 50 for $N = 20$, $N = 50$, and $N = 100$, respectively.

### A.3. Multi-Objective Knapsack Problem (MOKP)

In the Multi-Objective Knapsack Problem (MOKP), we consider $N$ items, each characterized by a weight $w_i$ and an $m$-dimensional profit vector $\mathbf{v}_i = (v_i^1, \ldots, v_i^m)$. The objective is to select a binary subset of items $\mathbf{x} \in \{0, 1\}^N$ that maximizes the accumulated profit for each objective, subject to the constraint that the total weight $\sum_{i=1}^{N} w_i x_i$ does not exceed the knapsack capacity $W$. To construct instances with inherent objective conflicts, we sample both the weights $w_i$ and profit vectors $\mathbf{v}_i$ independently and uniformly from the range $[0, 1]$. The knapsack capacity is typically set to $W = 12.5$ for standard problem sizes (i.e., $N \in \{50, 100\}$) and increased to $W = 25$ for larger instances ($N = 200$).

# B. Instance Augmentation

To boost the quality of the Pareto front during inference, we employ an instance augmentation strategy (Lin et al., 2022). This method exploits the geometric symmetry of Euclidean routing problems. For any node coordinate $(x, y)$, we can generate eight equivalent transformations via rotation and reflection, defined as $\{(x, y), (y, x), (x, 1 - y), (y, 1 - x), (1 - x, y), (1 - y, x), (1 - x, 1 - y), (1 - y, 1 - x)\}$. The number of augmented instances depends on the problem structure. For MOCVRP, applying these transformations yields 8 variants. Conversely, for MOTSP with $M$ objectives, transformations are applied independently to each objective to explore combinatorial variations, resulting in $8^M$ instances (64 for Bi-TSP and 512 for Tri-TSP).

# C. Hypervolume Indicator and Settings

We employ the Hypervolume (HV) indicator as the primary metric to assess the quality of the approximated Pareto front, as it simultaneously evaluates convergence and diversity. Formally, given a solution set $\mathcal{P} \subset \mathbb{R}^M$ and a reference point $\mathbf{r}$, the

HV value is defined as the Lebesgue measure $\Lambda(\cdot)$ of the union of hypercubes dominated by $\mathcal{P}$ and bounded by $\mathbf{r}$:

$$HV(\mathcal{P}, \mathbf{r}) = \Lambda \left( \bigcup_{\mathbf{y} \in \mathcal{P}} \{\mathbf{z} \mid \mathbf{y} \prec \mathbf{z} \prec \mathbf{r}\} \right). \tag{13}$$

To eliminate scale differences across objectives, we normalize all objective values to the range $[0, 1]$ before calculation. Specifically, for each problem instance, we determine the ideal point $\mathbf{z}^*$ and nadir point $\mathbf{z}^{nad}$ from the obtained solution set. The normalized value is computed as $\tilde{y}_m = (y_m - z_m^*)/(z_m^{nad} - z_m^*)$. For MOKP (maximization), values are inverted to fit the minimization context. The reference point $\mathbf{r}$ is set strictly according to the problem scale to ensure valid boundary evaluation. The detailed settings for reference points and ideal points used in our experiments are summarized in Table 5.

Table 5. Reference points ($r^*$) and ideal points ($z$) employed for HV calculation.

| PROBLEM | SIZE | $r^*$ | $z$ |
|---|---|---|---|
| | 20 | $(20, 20)$ | $(0, 0)$ |
| | 50 | $(35, 35)$ | $(0, 0)$ |
| BI-TSP | 100 | $(65, 65)$ | $(0, 0)$ |
| | 150 | $(85, 85)$ | $(0, 0)$ |
| | 200 | $(115, 115)$ | $(0, 0)$ |
| | 20 | $(30, 4)$ | $(0, 0)$ |
| BI-CVRP | 50 | $(45, 4)$ | $(0, 0)$ |
| | 100 | $(80, 4)$ | $(0, 0)$ |
| | 50 | $(5, 5)$ | $(30, 30)$ |
| BI-KP | 100 | $(20, 20)$ | $(50, 50)$ |
| | 200 | $(30, 30)$ | $(75, 75)$ |
| | 20 | $(20, 20, 20)$ | $(0, 0, 0)$ |
| TRI-TSP | 50 | $(35, 35, 35)$ | $(0, 0, 0)$ |
| | 100 | $(65, 65, 65)$ | $(0, 0, 0)$ |

# D. More Details of PMSA rchitecture

In this section, we provide a detailed elaboration of the encoder and decoder architectures in PMSA. While our base architecture draws inspiration from the conditional attention mechanism of WE-CA (Chen et al., 2025a), a critical distinction lies in our integration of explicit edge features to provide structural bias, making the model sensitive to preference-induced topological changes.

### D.1. Encoder

The encoder aims to map the input node features and preference vector into high-dimensional representations.

**Initialization.** Initially, the preference embedding $\mathbf{h}_\lambda^{(0)}$ and node embeddings $\mathbf{h}_i^{(0)}$ are obtained via linear projections of the input preference vector $\boldsymbol{\lambda}$ and raw node features $\mathbf{v}_i$:

$$\mathbf{h}_\lambda^{(0)} = \mathbf{W}_\lambda \boldsymbol{\lambda} + \mathbf{b}_\lambda, \quad \mathbf{h}_i^{(0)} = \mathbf{W}_v \mathbf{v}_i + \mathbf{b}_v, \quad \forall i \in \{1, \ldots, N\}. \tag{14}$$

where $\mathbf{W}_\lambda, \mathbf{b}_\lambda, \mathbf{W}_v, \mathbf{b}_v$ are trainable parameters. Additionally, we initialize explicit edge embeddings $\mathbf{e}_{ij}^{(0)}$ based on the pairwise distances between nodes.

First, the node embeddings conditioned on the weight embedding are produced by a feature-wise affine transformation (Perez et al., 2018), as follows:

$$\boldsymbol{\gamma} = \mathbf{W}_\gamma \mathbf{h}_\lambda^{(l-1)}, \quad \boldsymbol{\beta} = \mathbf{W}_\beta \mathbf{h}_\lambda^{(l-1)}. \tag{15}$$

$$\mathbf{h}_i'^{(l-1)} = \boldsymbol{\gamma} \odot \mathbf{h}_i^{(l-1)} + \boldsymbol{\beta}, \quad \forall i \in \{1, \ldots, N\}, \tag{16}$$

where $\odot$ denotes element-wise multiplication. This step aligns the node features with the specific objectives prioritized by $\boldsymbol{\lambda}$.

**Structural Attention with Edge Bias.** Unlike the standard MHA used in prior works, PMSA injects structural bias directly into the attention mechanism. Specifically, the edge features $\mathbf{e}_{ij}^{(l-1)}$ are projected into a preference-dependent bias term $\mathbf{B}^{(l-1)}(\lambda)$. Consequently, the preference and node embeddings are updated via the proposed structural attention (denoted as PMSA) and an Instance Normalization (IN) layer:

$$\hat{\mathbf{h}}_\lambda = \text{IN}\left(\mathbf{h}_\lambda^{(l-1)} + \text{PMSA}\left(\mathbf{h}_\lambda^{(l-1)}, \mathbf{H}_{ctx}, \mathbf{B}^{(l-1)}(\lambda)\right)\right). \tag{17}$$

$$\hat{\mathbf{h}}_i = \text{IN}\left(\mathbf{h}_i^{(l-1)} + \text{PMSA}\left(\mathbf{h}_i'^{(l-1)}, \mathbf{H}_{ctx}, \mathbf{B}^{(l-1)}(\lambda)\right)\right), \quad \forall i \in \{1, \ldots, N\}. \tag{18}$$

where $\mathbf{H}_{ctx} = \{\mathbf{h}_1'^{(l-1)}, \ldots, \mathbf{h}_N'^{(l-1)}\}$ represents the context set containing the conditioned node embeddings. Note that we exclude $\mathbf{h}_\lambda$ from $\mathbf{H}_{ctx}$ to avoid redundancy, as it already serves as the query in the weight update step.

Afterwards, a fully connected feed-forward sublayer and another Add & Norm sublayer are employed to yield the output embeddings $\mathbf{h}_\lambda^{(l)}$ and $\mathbf{h}_i^{(l)}$ for layer $l$.

### D.2. Decoder

The decoder generates the solution sequence $\pi$ auto-regressively. At each time step $t$, it leverages the final weight embedding $\mathbf{h}_\lambda^{(L)}$ and node embeddings $\mathbf{h}_i^{(L)}$ to produce a context query vector $\mathbf{h}_c$, the specific formulation of which captures the dynamic state of the optimization process. For MOTSP, $\mathbf{h}_c$ is formed by concatenating the embeddings of the first and last visited nodes, while for MOCVRP, it comprises the embedding of the last visited node augmented with the remaining vehicle capacity. In the case of MOKP, $\mathbf{h}_c$ integrates the global graph representation $\bar{\mathbf{h}} = \frac{1}{N+1}\sum_{i=0}^{N}\mathbf{h}_i$ with the residual knapsack capacity. Based on this context, we calculate a glimpse vector $\mathbf{q}_c$ via a standard Multi-Head Attention (MHA) mechanism over the encoder outputs to aggregate global graph information:

$$\mathbf{q}_c = \text{MHA}(\mathbf{h}_c, \{\mathbf{h}_\lambda^{(L)}, \mathbf{h}_1^{(L)}, \ldots, \mathbf{h}_N^{(L)}\}). \tag{19}$$

Subsequently, the glimpse vector $\mathbf{q}_c$ is enriched via a residual connection with the original context query $\mathbf{h}_c$ and further processed by a feed-forward network (FFN) to yield the final decoding query $\tilde{\mathbf{q}}_c$:

$$\mathbf{q}_c' = \mathbf{q}_c + \mathbf{h}_c, \tag{20}$$

$$\tilde{\mathbf{q}}_c = \mathbf{q}_c' + \text{FFN}(\mathbf{q}_c'). \tag{21}$$

This enhanced query vector is then employed to compute the attention scores for all candidate nodes. Specifically, the score $\alpha_j$ for each node $j$ is calculated using a single-head attention mechanism, where invalid nodes are explicitly masked to negative infinity:

$$\alpha_j = \begin{cases} -\infty, & \text{if node } j \text{ is masked,} \\ C \cdot \tanh\left(\dfrac{\tilde{\mathbf{q}}_c^\top \mathbf{W}_K \mathbf{h}_j^{(L)}}{\sqrt{d}}\right), & \text{otherwise.} \end{cases} \tag{22}$$

Here, the clipping constant $C$ is fixed at 10 to regulate the range of logits and prevent gradient saturation, following the protocol in (Kool et al., 2018). The final node selection probability $P_\theta(\pi_t | \pi_{1:t-1}, \lambda)$ is ultimately derived by applying the Softmax function over the computed scores $\alpha$, transforming them into a valid probability distribution.

## E. Training Algorithm Details

Algorithm 1 outlines the overall training procedure of PMSA. To equip the model with cross-scale generalization capabilities, we train a unified model by dynamically sampling instances from varying problem sizes $n$ during the training phase instead of the fixed scale.

## F. Experimental Results on Benchmark Instances

We validate the out-of-distribution generalization capability of PMSA through experiments on the KroAB dataset, as reported in Table 6.

---

**Algorithm 1** Unified Training Strategy for PMSA

---

1: **Input:** Iterations $K$, batch size $B$, size distribution $\mathcal{D}_{size}$, instance distribution $\mathcal{G}$, preference distribution $\Lambda$.
2: **Initialize:** Parameters $\theta$.
3: **for** $k = 1$ **to** $K$ **do**
4:     Sample problem scale $N \sim \mathcal{D}_{size}$ and preference vector $\boldsymbol{\lambda} \sim \Lambda$.
5:     Sample solutions $\pi_i^j \sim P_\theta(\cdot | \boldsymbol{\lambda}, s_i)$ for $i \in \{1, \ldots, B\}, j \in \{1, \ldots, N\}$.
6:     $b_i \leftarrow \frac{1}{N} \sum_{j=1}^{N} L(\pi_i^j | \boldsymbol{\lambda}, s_i)$.
7:     $\nabla \mathcal{J} \leftarrow \frac{1}{B \cdot N} \sum_{i=1}^{B} \sum_{j=1}^{N} \left[ (L(\pi_i^j | \boldsymbol{\lambda}, s_i) - b_i) \nabla_\theta \ln P_\theta(\pi_i^j | \boldsymbol{\lambda}, s_i) \right]$
8:     $\theta \leftarrow \text{Adam}(\theta, \nabla \mathcal{J})$
9: **end for**
10: **Output:** Optimized $\theta$.

---

*Table 6.* Performance on KroAB Benchmark Instances.

| METHOD | KROAB100 HV (↑) | GAP (↓) | TIME (↓) | KROAB150 HV (↑) | GAP (↓) | TIME (↓) | KROAB200 HV (↑) | GAP (↓) | TIME (↓) |
|---|---|---|---|---|---|---|---|---|---|
| WS-LKH | **0.7022** | **-0.20%** | 2.3M | **0.7017** | **-0.30%** | 4.0M | **0.7430** | **-0.41%** | 5.6M |
| MOEA/D | 0.6836 | 2.45% | 5.8M | 0.6710 | 4.09% | 7.1M | 0.7106 | 3.97% | 7.3M |
| NSGA-II | 0.6676 | 4.74% | 7.0M | 0.6552 | 6.35% | 7.9M | 0.7011 | 5.26% | 8.4M |
| MOGLS | 0.6817 | 2.73% | 52M | 0.6671 | 4.65% | 1.3H | 0.7083 | 4.28% | 1.6H |
| PPLS/D-C | 0.6785 | 3.18% | 38M | 0.6659 | 4.82% | 1.4H | 0.7100 | 4.05% | 3.8H |
| DRL-MOA | 0.6903 | 1.50% | 10s | 0.6794 | 2.89% | 18s | 0.7185 | 2.91% | 23s |
| MDRL | 0.6881 | 1.81% | 10s | 0.6831 | 2.36% | 17s | 0.7209 | 2.58% | 23s |
| EMNH | 0.6900 | 1.54% | 9s | 0.6832 | 2.34% | 16s | 0.7217 | 2.47% | 23s |
| PMOCO | 0.6878 | 1.86% | 9s | 0.6819 | 2.53% | 17s | 0.7193 | 2.80% | 23s |
| CNH | 0.6947 | 0.87% | 16s | 0.6892 | 1.49% | 19s | 0.7250 | 2.03% | 22s |
| WE-CA | 0.6948 | 0.86% | 9s | 0.6924 | 1.03% | 19s | 0.7317 | 1.12% | 26s |
| **PMSA** | 0.6996 | 0.17% | 10s | 0.6976 | 0.29% | 21s | 0.7372 | 0.38% | 32s |
| MDRL-AUG | 0.6950 | 0.83% | 13s | 0.6890 | 1.52% | 19s | 0.7261 | 1.88% | 28s |
| EMNH-AUG | 0.6958 | 0.71% | 12s | 0.6892 | 1.49% | 18s | 0.7270 | 1.76% | 27s |
| PMOCO-AUG | 0.6937 | 1.01% | 12s | 0.6886 | 1.57% | 19s | 0.7251 | 2.01% | 32s |
| CNH-AUG | 0.6980 | 0.40% | 17s | 0.6938 | 0.83% | 26s | 0.7303 | 1.31% | 37s |
| WE-CA-AUG | 0.6990 | 0.26% | 14s | 0.6957 | 0.56% | 23s | 0.7349 | 0.69% | 39s |
| **PMSA-AUG** | 0.7008 | 0.00% | 17s | 0.6996 | 0.00% | 26s | 0.7400 | 0.00% | 49s |

## G. Comparison on the Decomposition Method

The decomposition strategy transforms the original multi-objective problem into a series of scalar optimization subproblems. In this work, we investigate two mainstream scalarization functions: the Weighted Sum (WS) and the Tchebycheff (TCH) approach.

The Weighted Sum method aggregates objectives through a linear combination, offering computational efficiency and gradient simplicity. It is formulated as:

$$\min_{\mathbf{x} \in \mathcal{X}} g^{ws}(\mathbf{x} | \boldsymbol{\lambda}) = \sum_{m=1}^{M} \lambda_m f_m(\mathbf{x}). \tag{23}$$

The Tchebycheff scalarization is employed to handle non-convex geometries. It minimizes the maximum weighted deviation from an ideal reference point $\mathbf{z}^*$:

$$\min_{\mathbf{x} \in \mathcal{X}} g^{tch}(\mathbf{x} | \boldsymbol{\lambda}) = \max_{1 \le m \le M} \left\{ \lambda_m | f_m(\mathbf{x}) - z_m^* | \right\}, \tag{24}$$

where $\mathbf{z}^* = (z_1^*, \ldots, z_M^*)$ represents the utopian point, satisfying $z_m^* < \min_{\mathbf{x}} f_m(\mathbf{x})$. Although TCH provides better theoretical convergence on complex fronts, it introduces non-smoothness into the optimization landscape.

As shown in Table 7, we compared PMSA combined with Weighted Sum (WS) and Tchebycheff (TCH) decomposition strategies. PMSA (TCH) excels on small-scale instances, primarily due to its theoretical capability to explore non-convex regions inherent in discrete, small-scale solution spaces. Conversely, PMSA (WS) becomes dominant as the problem scale increases. This shift stems from the tendency of large-scale Pareto fronts to approximate a convex shape, coupled with the fact that WS provides denser gradient signals that construct a smoother optimization landscape for stable convergence.

*Table 7.* Performance comparison on different scalarization methods.

| PROBLEM SIZE | WE-CA (WS) | PMSA (WS) | PMSA (TCH) |
|---|---|---|---|
| BI-TSP20 | 0.6270 | 0.6272 | **0.6284** |
| BI-TSP50 | 0.6392 | **0.6412** | **0.6412** |
| BI-TSP100 | 0.7034 | **0.7070** | 0.7057 |
| BI-TSP150 | 0.7008 | **0.7062** | 0.7040 |
| BI-TSP200 | 0.7346 | **0.7408** | 0.7387 |

## H. GPU Memory Usage Analysis

Table 8 presents the GPU memory consumption for all models trained on instances with size $N = 50$. While PMSA incurs a moderate increase in memory overhead compared to baselines due to the joint processing of edge features, its peak usage remains efficiently low ,ensuring computational feasibility on standard hardware.

*Table 8.* Comparison of GPU Memory Usage.

| PROBLEM SIZE | WE-CA | CNH | PMSA |
|---|---|---|---|
| BI-TSP50 | 1051 MB | 1077 MB | 2084 MB |
| MOCVRP50 | 1790 MB | 1501 MB | 3058 MB |
| BI-KP50 | 916 MB | 916 MB | 1890 MB |
| TRI-TSP50 | 1052 MB | 1077 MB | 2086 MB |

## I. Summary of MOCOP Solvers

The characteristics of decomposition-based neural MOCO methods are summarized in Table 9. PMSA outperforms advanced neural baselines while maintaining a minor parameter footprint, effectively maximizing both parameter efficiency and solution quality.

*Table 9.* Summary of MOCOP Solvers

| Method | Techniques | Flexibility | # Params |
|---|---|---|---|
| DRL-MOA | Transfer learning | Multi-model | 133.37M |
| MDRL | Meta learning | Multi-model | 133.37M |
| EMNH | Meta learning | Multi-model | 133.37M |
| PMOCO | Hypernetwork | Single-model | 1.50M |
| CNH | Size-aware decoder | Single-model | 1.63M |
| WE-CA | Condition Attention | Single-model | 1.27M |
| PMSA | Node & Edge input | Single-model | 1.60M |

