# OpenReview forum: "Preference-Modulated Structural Attention for Multi-Objective Combinatorial Optimization"
_ICML.cc/2026/Conference — ICML 2026 regular_

### Official Review · Reviewer_QSen · 2026-03-05

**Soundness:** 3
**Presentation:** 3
**Significance:** 3
**Originality:** 3
**Overall Recommendation:** 4
**Confidence:** 3

**Summary:**

The authors propose a lightweight neural solver for multi-objective combinatorial optimization (MOCO). Their method improves how problem instances are represented by combining node and edge information within an attention-based model. They introduced a preference-modulated structural attention mechanism, where edge features are directly injected into the attention computation as structural biases. This allows the model to capture preference-specific relationships without adding significant computational cost. In addition, the method includes a node-guided edge aggregation strategy, where node information dynamically updates edge representations during encoding. This helps the model better recognize sub-problem structures and relationships in the graph.

**Compliance With Llm Reviewing Policy:**

Affirmed.

**Final Justification:**

The authors addressed my comments well.

**Key Questions For Authors:**

The edge features for the knapsack problem are constructed using a heuristic based on value-to-weight ratios. Could the authors justify this design choice and discuss whether alternative pairwise formulations were explored?

Could the authors further clarify how the Softmax operation in Eq. (9) is applied (e.g., row-wise or across attention heads), and how this normalization influences the edge aggregation process?

**Limitations:**

YES

**Strengths And Weaknesses:**

Strength

The paper addresses an important limitation in recent MOCO neural solvers that mainly rely on node-based representations by utilizing edge features in a preference-aware manner. The proposed design effectively captures complementary structural information from both node and edge features while maintaining low computational complexity.

The paper is well structured and easy to follow.

The evaluation is comprehensive and well designed. Experiments cover multiple MOCO tasks (Bi-/Tri-TSP, Bi-CVRP, and Bi-KP) across different problem sizes and include out-of-distribution generalization tests.

Weaknesses

While the authors note that predefined preference weights may limit flexibility when handling irregular Pareto fronts, the framework also relies on weighted-sum scalarization, which is known to have limited ability to recover solutions in non-convex regions of the Pareto front. It would be helpful if the authors could discuss how this limitation may affect Pareto front coverage in the proposed framework. Exploring alternative scalarization methods or adaptive preference sampling strategies could further improve solution diversity.


The article contains several minor spacing errors, for example in line 15 there should be a space here “However,existing” and in line 26 also “preferences,and”. The authors are advised to carefully proofread the manuscript and revise similar issues throughout the paper.

---

> ### Author Rebuttal · Authors · 2026-03-30
>
> We genuinely value and appreciate the time and effort the reviewer dedicated to evaluating our paper and offering insightful feedback.
> ### Response to the Weakness 1:
>
> We sincerely thank the reviewer for the insightful comments regarding the limitations of Weighted-sum scalarization in non-convex regions. We acknowledge that while WS is widely used in Neural Combinatorial Optimization for its efficiency, it may struggle to recover solutions in the concave portions of a Pareto front. However, we wish to clarify that the PMSA architecture, specifically our Edge-stream and dual-stream attention mechanisms, is fundamentally decoupled from the choice of scalarization function, meaning PMSA can be seamlessly integrated with other decomposition methods like Tchebycheff to capture solutions in non-convex regions. To evaluate this, we conducted experiments on Bi-TSP instances with clustered node distributions, which typically exhibit more complex front geometries. As shown in Table 1, PMSA remains compatible across both WS and TCH variants, with TCH showing a slight advantage as the problem scale increases. Furthermore, we observed that increasing the density of preference vectors significantly enhances the hypervolume  performance and front coverage, as detailed in Table 2, where finer preference sampling allows the neural network to explore the objective space more thoroughly. In conclusion, we agree that selecting appropriate decomposition methods and implementing adaptive preference sampling strategies based on specific problem distributions are critical for further improving diversity, and we consider these as promising directions for our future research.
>
> **Table 1. Performance Comparison of WS and TCH Scalarizations on Clustered Bi-TSP**
> | Model Setting | Bi-TSP20 | Bi-TSP50 | Bi-TSP100 | Bi-TSP150 | Bi-TSP200 |
> | :--- | :---: | :---: | :---: | :---: | :---: |
> | PMSA (WS) | 0.7748 | 0.8230 | 0.8677 | 0.8702 | 0.7387 |
> | PMSA (TCH) | **0.7748** | **0.8228** | **0.8683** | **0.8720** | **0.7408** |
>
> **Table 2. Impact of Preference Vector Density (N) on Tri-TSP Performance**
> | Model Setting | Tri-TSP20 | Tri-TSP50 | Tri-TSP100 |
> | :--- | :---: | :---: | :---: |
> | PMSA (N=105) | 0.4712 | 0.4430 | 0.5044 |
> | PMSA (N=1035) | 0.4747 | 0.4562 | 0.5236 |
> | PMSA (N=10011) | **0.4759** | **0.4593** | **0.5285** |
> ### Response to the Weakness 2:
> We sincerely apologize for the formatting oversights. We will carefully review and proofread the entire manuscript to correct all such spacing errors.
> ### Response to Question 1:
>
> As shown in Table 3, the proposed ratio-based formulation consistently yields superior results on large-scale instances. Our choice to use a value-to-weight ratio heuristic for KP edge features is grounded in classical greedy criteria, providing the Edge-stream with high-quality structural priors that reflect competitive relationships between items. As shown in our ablation study, while simple additive fusion (PMSA-add) already outperforms the baseline, the proposed PMSA-ratio formulation achieves the best performance. This confirms that incorporating domain-specific priors into the Edge-stream effectively guides the model through increasingly sparse Pareto fronts compared to basic fusion methods.
>
> **Table 3. Ablation Study on KP Edge Feature Formulation (HV)**
> | Instance | WE-CA (Baseline) | PMSA-add | **PMSA-ratio ** |
> | :--- | :---: | :---: | :---: |
> | Bi-KP (N=500) | 0.3105 | 0.3223 | **0.3252** |
> | Bi-KP (N=750) | 0.2496 | 0.2684 | **0.2704** |
> ### Response to Question 2:
>
> As with many neural-based multi-objective combinatorial optimization  methods such as P-MOCO[1] and WE-CA[2], our framework also utilizes the Softmax operation as a core normalization component. In our implementation, the Softmax operation is applied row-wise and independently for each attention head, typically acting on the last dimension of the attention tensor. This normalization ensures that for each query node i, the attention weights across all neighbors sum to 1, effectively transforming the Edge-stream enhanced scores into a relative importance distribution. This normalization introduces a competitive mechanism into the edge aggregation. If a specific edge feature is more congruent with the current preference, the Softmax operation exponentially amplifies its weight, allowing it to play a dominant role in updating node representations or subsequent edge-stream layers. This mechanism allows the model to dynamically filter the most critical structural connections relevant to the current preference vector while maintaining numerical stability across deep layers.
>
> [1] Pareto set learning for neural multi-objective combinatorial optimization, in ICLR 2022.
>
> [2] Rethinking neural multi-objective combinatorial optimization via neat weight embedding,  in ICLR 2025.

---

> > ### Author Rebuttal · Reviewer_QSen · 2026-04-02
> >
> > I thank the authors for the detailed responses. I updated my scores accordingly.

---

> > > ### Author Response · Authors · 2026-04-03
> > >
> > > Thank you for your kind feedback and for updating the score. We truly appreciate your thoughtful comments, which helped us refine and strengthen our work.

---

### Official Review · Reviewer_4N2V · 2026-03-09

**Soundness:** 3
**Presentation:** 2
**Significance:** 3
**Originality:** 3
**Overall Recommendation:** 5
**Confidence:** 4

**Summary:**

This paper proposes a new neural network-based algorithm for multi-objective combinatorial optimization, which is called the preference-modulated structural attention model (PMSA). The basic idea of the proposed algorithm is to efficiently utilize not only node features (which has already been used in existing algorithms) but also edge features (which is a new idea in this paper). High performance of the proposed algorithm is demonstrated through computational experiments on three types of test problems (TSP, CVRP, and KP) in comparison with various algorithms including both evolutionary approaches and neural network-based approaches.

**Compliance With Llm Reviewing Policy:**

Affirmed.

**Final Justification:**

The rebuttal fully addressed my concerns. I have changed my evaluation from “4: Weak accept” to “5: Accept”. This is because the rebuttal clearly shows that the proposed algorithm has high performance on multi-objective combinatorial optimization problems including large-scale problems.

**Key Questions For Authors:**

I will increase my evaluation if some questions are clearly answered in a convincing manner.

(i) I think that the presentation quality can be significantly improved during the rebuttal period.

(ii) Did you examine the scalability of the proposed algorithm with respect to the number of objectives (e.g., 5-objective problems and 10-objective problems used in the evolutionary multi-objective optimization field)?  Could you please include some comments ?

(iii) Did you examine the scalability of the proposed algorithm with respect to the problem size (e.g., 500-item and 750-item knapsack problems used in the evolutionary multi-objective optimization field)?  Could you please include some comments ?

(iv) Could you please include information about the training time, and justify the efficiency of the proposed algorithm in application scenarios ?  It seems that the efficiency of the proposed algorithm (and many other neural network-based approaches) depends on whether the trained model is used just once or many times. For example, if the training model is used for delivery problems by many delivery persons every day for several years, it seems that the long training time can be ignored. On the contrary, if the training model is used just once for a layout problem in a special facility, it seems that the comparison should be made based on the total computation time of each algorithm.

(v) Is it easy to apply the proposed algorithm to other multi-objective combinatorial optimization problems ?  As pointed out in this paper, one difficulty of traditional non-learnable solvers is that they need tailored operators to generate new solutions.  Different combinatorial optimization problems usually need different tailored operators, the performance of traditional non-learnable solvers heavily depends on the quality of those tailored operators, and the design of good tailored operators is not always easy (which usually need deep knowledge of domain experts). If it is clearly explained in a convincing manner why the proposed algorithm (and neural network-based approach in general) can be easily applied to various combinatorial optimization problems, the significance of this paper can be highly evaluated.

**Limitations:**

Yes.

**Strengths And Weaknesses:**

Soundness (Strengths):

- High performance of the proposed algorithm is demonstrated through computational experiments on three different types of multi-objective combinatorial optimization problems (TSP, CVRP, and KP) in comparison with various algorithms.

- Usefulness of the two main components in the proposed algorithms (i.e., the preference injection operation and the dynamic edge aggregation mechanism) is also clearly demonstrated through computational experiments.

- High generalization ability of the proposed algorithm is also clearly demonstrated using test problems with different size from the training examples.

Soundness (Weaknesses):

- The problem size of all test problems is small (e.g., 20, 50, 100, 150, 200 cities, and 20, 50, 100 items). Thus, the scalability of the proposed algorithm is not clear with respect to the problem size. In the EMO field, much larger test problems have been used in many studies.

- All test problems have only two objectives. Thus, the scalability of the proposed algorithm is not clear with respect to the number of objectives. In the EMO field, many-objective problems with four or more objectives (e.g., five objectives, ten objectives) have been used in many studies.

- Short computation time of the proposed algorithm is stressed in comparison with traditional non-learnable solvers as follows: “traditional non-learnable solvers require extensive computation time (e.g., WS-LKH takes 6 hours for Bi-TSP), PMSA achieves an optimality gap of 0.28% in just 38 seconds, and reaches a 0.1% gap within 22 minutes with instance augmentation.” I think that this short computation time does not include the training time, which may give somewhat unfair impression about the computation time of each algorithm.

Presentation (Strengths):

- It seems that the computational experiments and experimental results are well explained and shown.

Presentation (Weaknesses):

- This paper includes a huge number of minor presentation issues. The authors need to carefully read the paper again and again. The following are some examples.

- Some explanations are difficult to understand. For example, in the following sentence in the abstract, it is difficult to understand what “node-centric representations” and “complementary representations provided by edge features” mean: “However, existing methods typically rely exclusively on node-centric representations, failing to capture the complementary representations provided by edge features for problem instances, resulting in a persistent optimality gap.” This is because the previous sentence (i.e., the first sentence of the abstract) does not mention that “decomposition-based approaches” in this paper are not for evolutionary multi-objective optimization algorithms (e.g., MOEA/D, MOEA/DD, NSGA-III) but for neural network-based multi-objective optimization algorithms.

- More careful explanations may be needed for related works and compared algorithms. For example, dominance-based and decomposition-base MOEAs are explained as “the two most prominent paradigms”. However, there is a consensus among most EMO researchers that dominance-based, decomposition-based, and indicator-based MOEAs are the three main paradigms. The choice of representative papers for each paradigm is not standard (i.e., it seems to be biased): “dominance-based MOEAs (Deb et al., 2002; Deng et al., 2022) and decomposition-based MOEAs (Zhou et al., 2012; Hu et al., 2024).” Based on the popularity of each algorithm, NSGA-II and SPEA2 are usually used for dominance-based MOEAs, and MOEA/D and NSGA-III are usually used for decomposition-based MOEAs. Handing of irregular Pareto fronts is mentioned as the limitation of the current approach and a future research topic. This issue has already been discussed in the EMO field.  As a compared algorithm, “MOGLS” is explained as follows: “MOGLS (4,000 iterations with 100 local search steps) (Jaszkiewicz, 2002)”. Some additional explanations (e.g., as a footnote) may be needed for MOGLS since two different MOGLS algorithms are well known in the EMO community, which were proposed in the following papers: [1] Jaszkiewicz, A., Genetic local search for multi-objective combinatorial optimization. European journal of operational research, 137(1): 50–71, 2002, and [2] Ishibuchi H., Murata, T., A multi-objective genetic local search algorithm and its application to flowshop scheduling. IEEE Trans. on Systems, Man, and Cybernetics - Part C: Applications and Reviews, 28 (3): 392-403, 1998.

- This paper needs careful proofreading. The following are some examples, which are related to the use of a space: “However,existing methods”, “To address this , we propose”, “two streams,exact
methods and heuristic algorithms.”, “intractable for large-scale instances(Ehrgott et al., 2016;”, “particularly Multi-Objective Evolutionary Algorithms (MOEAs)(Tian et al., 2021)have”, “Hu et al., 2024) . Despite”, “variance,For more details see Appendix 1.”


Significance (Strengths):

- A new algorithm is proposed, and its high performance is clearly demonstrated.

Significance (Weaknesses):

- The scalability of  the proposed algorithm is not examined with respect to the problem size and the number of objectives.

- Performance improvement by the proposed algorithm from the existing algorithms looks very small (e.g., in Figure 2). Practical significance of such a small performance improvement is not clear.


Originality (Strengths):

- It seems that the proposed algorithm has high originality in the neural network-based multi-objective combinatorial optimization field.

- It seems that the neural network-based multi-objective combinatorial optimization field is a new research field.


Originality (Weaknesses):

- The originality of the proposed approach is not very clear in comparison with the related studies for single-objective combinatorial optimization. In the decomposition-based approach, a multi-objective combinatorial optimization problem is solved by solving a number of single-objective combinatorial optimization problems with different weight (preference) vectors. Thus, in principle, any single-objective approach can be used to solve multi-objective combinatorial optimization problems.

---

> ### Author Rebuttal · Authors · 2026-03-30
>
> We appreciate your valuable comments.
> ### Response to Question 1:
>
> We appreciate the reviewer’s professional corrections and will revise the manuscript to reflect a more rigorous taxonomy and update milestone references to align with EMO conventions. We also clarified the MOGLS baseline (Jaszkiewicz, 2002) and refined the discussion on irregular Pareto fronts to highlight our neural architecture's potential. Finally, we will conduct a thorough proofreading to correct all formatting errors and enhance scholarly rigor.
> ### Response to Question 2:
>
> Thank you for the question on many-objective optimization. To respect space limits, we have detailed our response and included many-objective test results in our reply to Question 1 for Reviewer 9AAF.
> ### Response to Question 3:
>
> To demonstrate scalability, we conducted additional experiments on large-scale benchmarks, including Bi-KP (up to N=750) and Bi-TSP (up to N=1000). As shown in Table 1, the performance gap between PMSA and the baseline widens significantly as problem size increases. This confirms that PMSA provide robust structural priors that remain effective for larger-scale problems, with the performance improvement becoming more pronounced as the scale grows.
>
> Table 1. Scalability Results on Large-scale TSP and KP
> | Problem Instance | WE-CA | PMSA  |
> | :--- | :---: | :---: |
> | Bi-TSP (N=300)| 0.6081 | **0.6190** |
> | Bi-TSP (N=500) | 0.3501 | **0.3923** |
> | Bi-TSP (N=1000) | 0.2529 | **0.3207** |
> | Bi-KP (N=500) | 0.3105 | **0.3252** |
> | Bi-KP (N=750) | 0.2496 | **0.2704** |
>
> ### Response to Question 4:
>
> As shown in Table 2, the training durations for several representative problem settings are recorded to provide a clear benchmark of the offline computational investment.We appreciate the reviewer’s perspective on the trade-off between training investment and application frequency. For high-frequency tasks like daily logistics, PMSA’s one-time training cost is negligible when amortized over thousands of inferences, where it vastly outperforms traditional meta-heuristics in decision speed. For low-frequency, customized tasks like facility layout, PMSA offers superior engineering efficiency; its POMO-based architecture allows complex constraints to be integrated via simple logic masks in a few lines of code, eliminating the need for experts to manually design specialized operators. Furthermore, as demonstrated by recent works like MTL-POMO[1] and MVMoE[2], this framework possesses strong zero-shot generalization, enabling the pre-trained model to be deployed directly or rapidly fine-tuned for new constraints. Thus, PMSA provides significant practical value by reducing both long-term computational overhead and the human-expert effort required for industrial deployment.
>
> Table 2. Training Time for Different Multi-Objective Combinatorial Problems
> | Task Setting | Bi-TSP50 | Bi-CVRP50 | Tri-TSP50 | Bi-KP50 |
> | :--- | :---: | :---: | :---: | :---: |
> | **Training Time** | 9.2h | 12.5h | 9.4h | 8.3h |
>
>
>
> ### Response to Question 5:
>
> We appreciate the reviewer’s constructive feedback on the adaptability of PMSA. A major limitation of traditional non-learning solvers is their heavy reliance on expert-designed operators, which must be specifically tailored for each problem's unique constraints. In contrast, PMSA replaces these manual heuristics with a learned policy and a flexible masking mechanism.
>
> To demonstrate this ease of application, we extended PMSA to the Multi-Objective Capacitated Vehicle Routing Problem with Time Windows. In this problem, each customer node is associated with a time window [ei, li] and a service time si. A vehicle must start service within the period ei to li, and all vehicles must return to the depot. This study considers three objectives: minimizing total travel distance, minimizing the number of routes, and minimizing the average route length per served customer.
>
> The adaptation to MOCVRPTW was highly efficient: we updated the input features to include time window coordinates and modified the logic masks to ensure feasibility by masking nodes that would violate time or capacity limits. As shown in Table 3, PMSA consistently outperforms the baseline across all instances without requiring any problem-specific operator engineering. By delegating strategy discovery to the neural network and handling constraints through masks, PMSA offers a highly generalized framework for diverse combinatorial optimization tasks.
>
> Table 3. Performance Comparison on MOCVRPTW Instances
> | Problem| WE-CA  | PMSA |
> | :--- | :---: | :---: |
> | MOCVRPTW20 | 0.4099 | **0.4126** |
> | MOCVRPTW50 | 0.4125 | **0.4177** |
> | MOCVRPTW100 | 0.3988 | **0.4058** |
>
> [1]Multi-task learning for routing problem with cross-problem zero-shot generalization.in KDD2024.
>
> [2]MVMoE: Multi-Task Vehicle Routing Solver with Mixture-of-Experts. in ICML 2024.

---

> > ### Author Rebuttal · Reviewer_4N2V · 2026-04-01
> >
> > Authors' responses to my questions are clear and convincing (except for the presentation quality improvement). I have updated my "Overall Recommendation".
> >
> > Please try to improve the presentation quality as much as possible, and also choose/include appropriate references.

---

> > > ### Author Response · Authors · 2026-04-03
> > >
> > > Thank you for your kind feedback and for updating the score. We truly appreciate your thoughtful comments, which helped us refine and strengthen our work.We will improve the presentation quality and choose the appropriate references as soon as possible..

---

### Official Review · Reviewer_9AAf · 2026-03-11

**Soundness:** 3
**Presentation:** 3
**Significance:** 3
**Originality:** 3
**Overall Recommendation:** 3
**Confidence:** 4

**Summary:**

This paper introduces PMSA (Preference-Modulated Structural Attention), a novel and lightweight neural solver designed for Multi-Objective Combinatorial Optimization (MOCO). While recent decomposition-based methods have made strides by treating MOCO problems as a series of scalarized sub-problems, they predominantly rely on node-centric representations. The authors argue that such approaches fail to capture critical local topological information provided by edge features, leading to a persistent optimality gap.

To address these limitations, PMSA deals node and edge features through two primary mechanisms: Instead of using heavy graph encoders that suffer from over-smoothing and high computational costs, the model directly injects preference-modulated edge features into the attention mechanism as explicit structural biases. This allows the model to perceive preference-specific distance characteristics and adjust information flow along edges based on the current optimization direction.

The researchers evaluated PMSA on standard MOCO benchmarks, including MOTSP, MOCVRP, and MOKP and achieve some better results.

**Compliance With Llm Reviewing Policy:**

Affirmed.

**Final Justification:**

keep my original score.

**Key Questions For Authors:**

1. While the study focuses on bi-objective problems, it would be insightful to understand how the proposed method scales as the number of objectives increases. Could the authors discuss the potential challenges or performance trends in many-objective scenarios?

2. The inclusion of MOEA/D and NSGA-II provides a solid traditional baseline. However, would it be possible to include a comparison with recent state-of-the-art neural solvers to further highlight the model's relative advantages?

3. Regarding the TSP 20 results, the performance gains appear relatively modest. Could the authors provide further context on whether these improvements are statistically significant or offer specific advantages in certain problem instances?The use of a 1% significance level for the Wilcoxon rank-sum test is quite stringent. Would the conclusions remain consistent at a more conventional 5% level, or was there a specific reason for choosing this stricter threshold?

4. The manuscript strives to build upon the foundation of the MO-POMO framework (Lin et al., 2022). The authors proceed to explore an important concept in refining this approach, but it would be beneficial to more clearly articulate the unique contributions that distinguish this work from subsequent follow-up studies.

5. Is there some relationship with the proposed key idea and the film method?


referece:
FiLM: Visual Reasoning with a General Conditioning Layer

**Limitations:**

See questions.

**Strengths And Weaknesses:**

1. Compared with hypernetwork methods, this method only rely on fewer parameters.

2. This paper consider MOO problems rather than SOO problems which are more diffuclt to tackle.

---

> ### Author Rebuttal · Authors · 2026-03-30
>
> Thank you for your insightful comments.
> ### Response to Question 1:
>
> PMSA’s effectiveness in many-objective scenarios stems from its ability to capture invariant structural constraints, such as node coordinates, which remain constant despite increasing objective dimensionality and solution sparsity. Stress tests on 5- and 10-objective TSP (see **Table 1**) confirm that PMSA achieves significant performance gains, demonstrating superior robustness over the baseline method. We acknowledge that many-objective combinatorial optimization still faces inherent challenges:
> * As the number of objectives increases, the number of required weight vectors grows exponentially, making it increasingly difficult to achieve full coverage of the entire Pareto front.
> * Existing sampling methods often struggle with non-uniform Pareto fronts. We will explore adaptive preference sampling to further enhance solution diversity in the revised version.
>
> **Table 1. HV Performance on Many-objective TSP**
> | Model Setting | m=5  | m=10  |
> | :--- | :---: | :---: |
> | WE-CA (N=20) | 0.3575 | 0.2263 |
> | PMSA (N=20) | **0.3581** | **0.2271** |
> | WE-CA (N=50) | 0.3268 | 0.2076 |
> | PMSA (N=50) | **0.3329** | **0.2111** |
> | WE-CA (N=100) | 0.3701 | 0.2597 |
> | PMSA (N=100) | **0.3785** | **0.2681** |
>
> ### Response to Question 2:
>
> We compared PMSA with the recent state-of-the-art method POCCO[1]. POCCO utilizes MoE-based routing to expand model capacity. In contrast, PMSA focuses on refining the internal relational modeling through an Edge-stream mechanism, which explicitly processes pairwise constraints like distances to prevent the model from losing structural focus. These two approaches are fundamentally orthogonal, and POCCO’s contributions could be integrated to further boost performance. Nevertheless, PMSA consistently outperforms POCCO across various benchmarks (see **Table 2**).
>
> **Table 2.  Hv Performance Comparison with POCCO on TSP Instances**
> | Model | Bi-TSP20  | Bi-TSP50 | Bi-TSP100 | Bi-TSP150 | Bi-TSP200 |
> | :--- | :---: | :---: | :---: | :---: | :---: |
> | WE-CA | 0.6270 | 0.6392 | 0.7034 | 0.7008 | 0.7346 |
> | POCCO-C | **0.6275** | 0.6411 | 0.7055 | 0.7033 | 0.7371 |
> | PMSA | 0.6272 | **0.6412** | **0.7070** | **0.7062** | **0.7408** |
>
> ### Response to Question 3:
>
> * The modest improvement at $N=20$ is because standard attention is generally sufficient for small-scale dependencies. However, as the scale increases, attention weights are inevitably spread thin over larger neighborhoods. Our Edge-stream mechanism counteracts this by acting as a structural anchor that modulates attention flow, reinforcing focus on high-priority neighbors while suppressing distant noise. By preventing attention from becoming overly dispersed, PMSA preserves structural integrity, leading to the significant performance leaps observed in larger and more complex instances.
> * To ensure the robustness of our findings, we utilized a strict 1% significance threshold. Our results (see **Table 3**) show that the improvements of PMSA are statistically robust, consistently surpassing the baselines beyond any stochastic fluctuations.
>
> **Table 3. Statistical Significance (p-values) across Different Scales**
> | Problem | N=20 (p-value) | N=50 (p-value) | N=100 (p-value) |
> | :--- | :---: | :---: | :---: |
> | **Bi-TSP** | 4.5e-3 | 8.5e-35 | 6.9e-35 |
> | **Bi-CVRP** | 7.2e-7 | 3.2e-22 | 1.6e-27 |
>
> ### Response to Question 4:
>
> PMSA shifts from a node-centric paradigm to a joint node-edge co-encoding framework to resolve the "structural dilution effect" in MOCO. We integrate two synergistic mechanisms:
> * Preference-Modulated Structural Attention, which injects edge attributes as explicit biases to ensure attention flow is strictly preference-conditioned.
> * Node-Guided Dynamic Edge Aggregation, where node context refines edge features to capture granular sub-problem structures. This dual-stream design prevents "attention dispersion," ensuring topological constraints remain sharp and providing robust control over multi-objective trade-offs compared to previous node-only architectures.
> ### Response to Question 5:
>
> While PMSA aligns with the philosophy of FiLM as a conditional modulation mechanism, we introduce key innovations specialized for MOCOPs. Unlike standard FiLM, which typically performs channel-wise scaling on node features or global states, our method extends modulation to the Edge-stream, enabling precise, preference-driven adjustments to relational constraints. Furthermore, instead of simple linear shifts, these modulated features are integrated as explicit structural biases within the attention mechanism. This evolves FiLM from a generic feature modulator into a structurally-aware framework that directly reshapes the attentional topology according to specific trade-offs, uniquely capturing complex geometric dependencies.
>
> [1] Preference-Driven Multi-Objective Combinatorial Optimization with Conditional Computation, in NeurIPS 2025

---

> > ### Author Rebuttal · Reviewer_9AAf · 2026-04-03
> >
> > Thanks for your rebuttal, however, the nolvey is a bit limited, and I keep my score.

---

> > > ### Author Response · Authors · 2026-04-04
> > >
> > > Thank you for your comments. We would like to further clarify our contributions for Neural MOCO by providing more details.
> > >
> > > Recent neural network-based MOCO methods[1][2][3] have mainly relied on node-centric representations. The key limitation is that these models have failed to capture the vital complementarity between global long-range dependencies and local topological structures, which leads to a persistent optimality gap. Recent SOCO methods[4][5] have confirmed this viewpoint, as we known, we are the first to propose an MOCO method that utilizes complementary representations of node features and edge features.
> > >
> > > Distinct from joint-input methods in SOCO, we move away from GCN-based edge feature extraction. This is because decomposition-based MOCO requires solving a large number of sub-problems simultaneously, where a heavy encoder would significantly exacerbate the total solving time. Therefore, we design PMSA, a lightweight architecture that directly injects preference-modulated edge features into the attention mechanism as a structural bias. Furthermore, we facilitate the participation of node features in the edge aggregation process. This allows the model to leverage edge features as a structural prior while enabling the dynamic update of preference-specific edge representations during encoding, thereby enhancing the model’s capability to perceive the structure of each sub-problem.
> > >
> > > Extensive evaluations demonstrate that our approach consistently achieves superior performance across diverse decomposition frameworks. PMSA exhibits remarkable robustness and generalization, maintaining a significant performance lead when scaling to larger problem sizes, many-objective scenarios, and non-uniform distributions. These results underscore the versatility and practical potential of our method in handling complex multi-objective challenges.
> > >
> > > [1] Pareto set learning for neural multi-objective combinatorial optimization, in ICLR 2022.
> > >
> > > [2] Rethinking neural multi-objective combinatorial optimization via neat weight embedding, in ICLR 2025.
> > >
> > > [3] Preference-Driven Multi-Objective Combinatorial Optimization with Conditional Computation, in NeurIPS 2025.
> > >
> > > [4] EFormer: An Effective Edge-based Transformer for Vehicle Routing Problems, IJCAI 2025.
> > >
> > > [5] UniteFormer: Unifying Node and Edge Modalities in Transformers for Vehicle Routing Problems, NeurIPS 2025.

---

### Official Review · Reviewer_9zgG · 2026-03-12

**Soundness:** 3
**Presentation:** 3
**Significance:** 2
**Originality:** 3
**Overall Recommendation:** 4
**Confidence:** 2

**Summary:**

This paper tackles Multi-Objective Combinatorial Optimization (MOCO) by training a single neural network to approximate the Pareto front across decomposed sub-problems. The key contribution is incorporating edge features modulated by preference vectors into the attention mechanism, whereas prior neural methods only used node features. Experiments on bi/tri-objective TSP, CVRP, and Knapsack show consistent improvements over state-of-the-art neural baselines with reasonable computational overhead.

**Compliance With Llm Reviewing Policy:**

Affirmed.

**Key Questions For Authors:**

1. How does the method perform on non-uniform or real-world instances where the structure differs significantly from synthetic benchmarks?
2. Given that GPU memory doubles, how does the approach scale when the number of sub-problems increases substantially?

**Limitations:**

Yes

**Strengths And Weaknesses:**

Strengths:

1. Addresses a clear and well-motivated gap, prior neural MOCO methods ignore edge features that are naturally available in combinatorial problems.
2. Clear experimental evaluation: multiple problem types, multiple scales, ablation studies, generalization to unseen sizes, statistical significance tests, and comparison against both neural and non-learnable baselines.
3. The method shows consistent improvements over strong baselines across all benchmarks, particularly substantial on larger instances and tri-objective problems.

Weaknesses:
1. Only evaluated on synthetic benchmarks with uniform distributions; no real-world validation, which limits practical significance.
2. GPU memory roughly doubles compared to baselines (Table 8) and inference time increases. In a decomposition setting with many simultaneous sub-problems, this could be a real bottleneck.
3. Improvements on small instances (N=20) are marginal; a discussion on when and why edge information actually helps would strengthen the paper.

---

> ### Author Rebuttal · Authors · 2026-03-30
>
> Thank you for your insightful comments.
> ### Response to Question 1:
>
> We clarify that our original study evaluated PMSA on the KroAB100 instance from TSPLIB (see Table 4 and Figure 2), which consists of standard benchmark nodes derived from actual geographic coordinates. To further examine the robustness of PMSA across diverse spatial patterns, we conducted additional experiments on Clustered, Exponential, and Mixed (50% Clustered, 50% Uniform) distributions. As shown in Table 1, PMSA maintains a consistent performance lead over the baseline across these structured scenarios, suggesting its potential for practical applications with non-uniform instances.
>
> **Table 1. Hypervolume (HV) Performance on Non-uniform Distributions (Bi-TSP)**
> | Distribution | Model | n=20 | n=50 | n=100 |
> | :--- | :--- | :---: | :---: | :---: |
> | **Clustered** | WE-CA | 0.7741 | 0.8204 | 0.8640 |
> | | PMSA  | **0.7748** | **0.8228** | **0.8683** |
> | **Exponential** | WE-CA | 0.6262 | 0.6398 | 0.7037 |
> | | PMSA  | **0.6276** | **0.6417** | **0.7060** |
> | **Mixed** | WE-CA | 0.6643 | 0.6959 | 0.7564 |
> | | PMSA  | **0.6652** | **0.6974** | **0.7587** |
>
> ### Response to Question 2:
>
> We acknowledge that PMSA involves a more complex architecture with higher GPU memory usage, but its parallel inference capability significantly mitigates time overhead. Most hypernetwork-based neural MOCO methods[1] learn weight-dependent model parameters, making it difficult to execute parallel solving across different preferences. By contrast, our method directly inputs the preference vector to learn specific representations, enabling efficient parallelization for a batch of subproblems. In our experiments on Tri-objective TSP (see Table 2), we used Das-Dennis decomposition to generate N subproblems (N=105, 1035, 10011). Parallel inference significantly curtails the total wall-clock time, making high-density preference sets practically feasible. While the speedup is not exactly linear to the number of subproblems due to hardware throughput and memory limits, the efficiency remains within an acceptable range. Although PMSA involves more computation than the baseline, the consistent improvement in Hypervolume justifies this trade-off for superior solution quality.
>
> **Table 2. Solving Time Comparison with Parallel Inference (Tri-TSP)**
> | Model Setting | HV  | Time  | Time (Parallel) |
> | :--- | :---: | :---: | :---: |
> | WE-CA (scale=100, N=105) | 0.4985 | 20s | 15s |
> | PMSA (scale=100, N=105) | **0.5044** | 36s | 28s |
> | WE-CA (scale=100, N=1035) | 0.5171 | 3.0m | 2.2m |
> | PMSA (scale=100, N=1035) | **0.5236** | 5.5m | 4.8m |
> | WE-CA (scale=100, N=10011) | 0.5225 | 29m | 23m |
> | PMSA (scale=100, N=10011) | **0.5285** | 44m | 32m |
>
> ### Response to Weakness 3:
>
> We appreciate the reviewer's observation regarding the performance variance across scales. The modest improvement on n=20 compared to the substantial gains on n=50 and n=100 is primarily due to the "dilution effect" inherent in standard attention mechanisms. In large-scale problems, attention weights are inevitably spread thin across a vast number of nodes, which "blurs" the model's focus on critical local structures. Our Edge-stream mechanism effectively counteracts this dispersion by acting as a structural anchor; it leverages edge-level features to modulate attention flow, naturally suppressing the influence of distant or irrelevant nodes while reinforcing the focus on high-priority neighbors. By preventing the attention from becoming overly distracted as the problem scale increases, PMSA preserves structural integrity, which becomes the decisive factor for the significant performance leaps observed in larger and higher-objective instances.
>
> [1] Pareto set learning for neural multiobjectative combotorial optimization, in ICLR 2022.

---

> > ### Author Rebuttal · Reviewer_9zgG · 2026-04-04
> >
> > The additional experiments on non-uniform distributions (Clustered, Exponential, Mixed) and the TSPLIB instance address my concern about synthetic-only evaluation. My concerns are adequately addressed. I maintain my score.

---

> > > ### Author Response · Authors · 2026-04-04
> > >
> > > Thank you for your feedback and for maintaining your score. We greatly appreciate your support and consideration.

---

### Decision · Program_Chairs · 2026-04-30

**Decision:**

Accept (regular)

**Comment:**

This paper addresses multi-objective combinatorial optimization by training a single neural network to approximate the Pareto front across decomposed sub-problems. The main innovation is incorporating edge features modulated by preference vectors into the attention mechanism, while prior neural methods only used node features. Experiments are performed to show consistent improvements over state-of-the-art neural baselines.

After the rebuttal, reviewers are generally positive. The problem to be addressed is well motivated. The experiments are comprehensive, showing consistent improvements over strong baselines.